**Review Article: A European Perspective on Wind and storm damage: From the**
**meteorological background to index-based approaches to assess Impacts**
Daniel Gliksman[1,2], Paul Averbeck[3], Nico Becker[4,5], Barry Gardiner[6,7], Valeri Goldberg[1], Jens
Grieger[4], Dörthe Handorf[8], Karsten Haustein[9,#], Alexia Karwat[10], Florian Knutzen[9], Hilke S.
Lentink[11], Rike Lorenz[4], Deborah Niermann[12], Joaquim G. Pinto[11], Ronald Queck[1], Astrid
Ziemann[1], and Christian L. E. Franzke[13,14]
[1] Technische Universität Dresden, Faculty of Environmental Sciences, Institute for
Hydrology and Meteorology, Chair of Meteorology, Pienner Str. 23, 01737 Tharandt
Germany
[2] Chair of Computational Landscape Ecology, Institute of Geography, Technische Universität
Dresden, Helmholtzstr. 10, 01069, Dresden, Germany
[3] iES Landau, Institute for Environmental Sciences, University of Koblenz-Landau, Fortstraße
7, 76829 Landau, Germany
[4] Freie Universität Berlin, Institut für Meteorologie, Berlin, Germany
[5] Hans-Ertel-Centre for Weather Research, Berlin, Germany
[6] Faculty of Environment and Natural Resources, Albert-Ludwigs University, Freiburg,
Germany.
[7] Institut Européen de la Forêt Cultivée, Cestas, France.
[8] Alfred Wegener Institute, Helmholtz Centre for Polar and Marine Research, Research
Department Potsdam, Telegrafenberg A45-ND-14473 Potsdam, Germany
[9] Climate Service Center Germany (GERICS), Helmholtz-Zentrum hereon, Fischertwiete 1,
20095 Hamburg, Germany
[10] Universität Hamburg, Meteorological Institute, Grindelberg 5, 20144 Hamburg, Germany
[11] Institute of Meteorology and Climate Research, Department of Tropospheric Research
(IMK-TRO), Karlsruhe Institute of Technology (KIT), Karlsruhe, Germany
[12] Deutscher Wetterdienst, Frankfurter Straße 135, 63067 Offenbach, Germany
[13] Center for Climate Physics, Institute for Basic Science, Busan, Republic of Korea
[14] Pusan National University, Busan, Republic of Korea
[#] Now at Institute for Meteorology, University of Leipzig, Leipzig, Germany
Corresponding author: christian.franzke@pusan.ac.kr
**Abstract** Wind and windstorms cause severe damage to natural and human-made
environments. Thus, wind-related risk assessment is vital for the preparation and mitigation of
calamities. However, the cascade of events leading to damage depends on many factors that
are environment-specific and the available methods to address wind-related damage often
require sophisticated analysis and specialization. Fortunately, simple indices and thresholds
are as effective as complex mechanistic models for many applications. Nonetheless, the
multitude of indices and thresholds available requires a careful selection process according to
the target sector. Here, we first provide a basic background on wind and storm formation and
characteristics, followed by a comprehensive collection of both indices and thresholds that can
be used to predict the occurrence and magnitude of wind and storm damage. We focused on
five key sectors: forests, urban areas, transport, agriculture, and wind-based energy
production. For each sector we described indices and thresholds relating to physical properties
such as topography and land cover but also to economic aspects (e.g. disruptions in
transportation or energy production). In the face of increased climatic variability, the promotion
of more effective analysis of wind and storm damage could reduce the impact on society and
the environment.

## 1. General introduction

Wind is a common feature of our day-to-day weather just like air temperature and precipitation.
Wind is per definition a sustained air movement in the atmosphere, which can range from still
conditions to extraordinary values, from very local to global spatial scales, and has a wide
range of temporal scales from seconds to decades. Winds can have both a beneficial and
detrimental effect on society, infrastructure, and agriculture. On one hand, storms, which have
very strong winds, can lead to considerable damage in infrastructure and forestry, e.g. storm
Kyrill in 2007 (Fink et al., 2009), contribute to widespread forest fires, e.g. Australia 2020 (van
Oldenborgh et al., 2021), or enhance evaporation, thus, drying out the soil (Bittelli et al., 2008).
We view damage as a disadvantageous change in the quantities, quality, or function of an
object. On the other hand, moderately strong winds can have positive effects on wind energy
production and cause a stronger mixing in the boundary layer (cancelling detrimental thermal
inversions to agriculture) or – in the case of nightly slope winds - alleviate summer heat
conditions in valleys and cities (Ganbat et al., 2015).

The damage associated with strong winds is primarily due to short-term wind gusts, and
leads to a substantial increase in wind speed (Brasseur, 2001). Wind gusts are sudden
increases in windspeed, which last typically less than 20 seconds, while strong winds refer to
sustained wind speed over longer time periods. Strong wind gusts often lead to uprooting or
breaking of trees, damage to crops in fields (Gardiner et al. 2016), lifting of roofs, and
damaging critical infrastructure like bridges and roads (Klawa and Ulbrich, 2003; Mitchell-
Wallace et al. 2017). In coastal areas, strong winds and wind gusts may lead to storm
surges and coastal flooding (Flather, 2001). The exact impacts of strong winds depend also
on other factors besides wind speed thresholds. For example, damage to forests depends
on many other factors like precipitation and topography (Gardiner, 2021). Thus, to predict
damage or identify areas at risk of wind or storm damage, indices are a vital tool in
assessing the likelihood and magnitude of damage in a given sector or environment. For
example, Merz et al. (2020) explore in their review the current state of knowledge on skillful
forecasts of impacts for many hazards, for which indices are very useful. With storm damage
we refer to damage, mainly to properties and forests, caused by severe wind storms, while
wind damage is more general and includes all adverse effects of wind, including storm
damage. We define risk as the likelihood here that wind causes some damage, and their
consequences and risk can be quantified as the function of hazard probability, exposure and
vulnerability (e.g. Kelman 2003; Hoeppe 2016; Franzke 2017).

For wind indices and wind impact models different wind parameters are in use. These are
often derived from modeled data like reanalysis datasets. While these model parameters are
strongly related to observed wind parameters, they are not the same and their definitions
cannot be used interchangeably. Since observational data is rare and it is more common to
work with modeled data the following parameter definitions focus on parameters derived
from models. It is often assumed that the maximum daily or hourly gust speed [m/s] at 10m
height relates strongest to damage. The WMO defines a wind gust as the maximum of the
wind averaged over 3 second intervals which is in most cases shorter than the model time
step. Thus, many models rely on parametrization for gust speed. For example, the ECMWF
Integrated Forecasting System deduces the magnitude of a gust within each time step from
the time-step-averaged surface stress, surface friction, wind shear and stability. Other
common parameters in use are daily or hourly mean or maximum wind speeds at 10m
height which express the mean or maximum values of all model time steps in an hour or a
day. The parametrized gust speed as well as mean wind speeds in a model grid cell can
deviate widely from local observations.

Indices can be used to predict damage caused directly by wind, or to quantify how the wind
modulates the damage caused by another process such as fire or drought. Furthermore, the
choice of indices depends also on land use as it influences the interaction between land
surfaces and the wind; tree species and forest structures can have considerable influence on
the damage probability (Gardiner, 2021). The understanding of wind, storm dynamics, and
the ability to predict the damage they cause, requires an interdisciplinary approach.
However, much of the relevant literature is in specialized journals. Here, we aim to bring
these different disciplines together to provide an interdisciplinary synthesis of the topic. To
bridge the gap between the different communities, within the ClimXtreme consortium, we
created a work group and invited specialists from outside the consortium to broaden our
research expertise. During regular joint meetings we identified the following sectors: forests,
urban areas, transport, agriculture, and energy as the most relevant terrestrial environments
that could be impacted by wind and storm damage. We focused on literature resources
stemming mainly from Europe, but in cases of relevance and to further expand the scope of
the review we also incorporated examples from other regions.
We provide a basic background on wind and storm formation and intra-seasonal variability in
section 2. Section 3 focuses on the interactions between wind and surface structures which
are prone to wind-damage. Section 4 focusses on wind- and storm-related indices and
thresholds.  In particular, we cover the following sectors: forests, urban areas, transport,
agriculture, and energy. Additionally, we discuss compound indices and thresholds used by
national weather services. Finally, in section 5 we provide an outlook and discuss open
research questions. Due to the location of the authors, we provide mainly a European
perspective on this topic, but believe our synthesis is more widely applicable.

**2. Wind and storm formation – mechanisms and concepts**
**2.1.    The general circulation and wind generation**
The general circulation of the atmosphere is driven by the differential heating of the Earth
(Held 2019); the equatorial regions receive more solar radiation than higher latitudes, while in
the polar regions the atmosphere is losing heat into space. This differential heating of the
Earths' surface causes pressure differences in the atmosphere. As a result, a pressure
gradient force acts on the air masses, leading to a movement from high to low pressure centers
to alleviate this pressure difference. Since the atmosphere moves toward an equilibrium, it
causes a meridional heat transport towards the poles through the atmosphere and ocean,
which takes place mainly through the movements of circulation systems and storms (Bjerknes
1922; Schultz et al. 2019; Ma et al. 2021).

Mid-latitude weather systems include both cyclones and anticyclones, but strong wind
situations are primarily associated with intense cyclones. The main paths that weather
systems and storms take, are called storm tracks (Hoskins and Valdez 1990; Blender et al.
1997; Chang et al. 2002; Ulbrich et al. 2009). Storm tracks form over the major ocean basins
of the Northern and Southern hemispheres and are closely related to atmospheric jet-streams,
which are areas of maximum upper-level wind speed and determine the areas that are prone
to storms as discussed below in section 2.4. These regimes set the propensity with which
weather systems take a more poleward or equatorward path on intra-seasonal time scales,
thus offering potential predictability.

In its most basic form, atmospheric jet-streams (Feldstein and Franzke 2017) are a product of
the pressure gradient force, induced by the above-mentioned latitudinal air temperature
gradients, and the Coriolis force. For large-scale movements in the atmosphere, the wind is
diverted to the right (left) in the northern (southern) hemisphere due to the Coriolis force. The
resulting winds in the free atmosphere, above the boundary layer, blow parallel to lines of
equal pressure, in a balance between the pressure gradient and the Coriolis force; also called
geostrophic wind. The strength of the dominant westerly winds over Western Europe is
determined by the pressure difference between the subpolar and subtropical regions over the
eastern North Atlantic. The stronger the pressure difference, the stronger the mid-latitude
westerlies.

Under hypothetical unperturbed conditions, the bands of maximum wind speed sit at 30° and
60° latitude in either hemisphere at upper levels of the troposphere, due to surface friction.
However, differential diabatic heating over land and the ocean, or orographic surface features,
such as mountains, do perturb the jet-stream in multiple ways. As a result, in the extra-tropics
of the northern hemisphere the jet-stream is commonly split into a subtropical and mid-
latitudinal branch. While the former is mainly driven by angular momentum transport by the
thermally direct Hadley circulation (Held and Hou 1980), the latter is primarily driven by the
eddy momentum flux convergence provided by short waves that form in regions of enhanced
baroclinicity (Held 1975). Accordingly, the mid-latitudinal jet-stream is referred to as an eddy-
driven or polar jet-stream due to its proximity to polar latitudes.

In the atmosphere unstable conditions are needed for weather systems to form (Holton and
Hakim 2012). So-called baroclinically unstable conditions occur where we find strong
horizontal and vertical air temperature gradients. For example, the North Atlantic is an ideal
source region for baroclinically unstable conditions as very cold polar air is advected over
moderately warm ocean waters, leading to excessive air temperature gradients and, thus,
pressure gradients, which – under the influence of the Coriolis force – generate enhanced
baroclinicity.

In the boundary layer, the pressure gradient and Coriolis forces are not in balance, because
the surface characteristics, local conditions, vertical stability, and other effects play crucial
roles in modifying the winds. Under the influence of surface friction, the air movements are not
parallel to the lines of equal pressure but have a tangential component from high to low
pressure centers. On the regional to local scale, wind systems like the land-sea-breeze, and
mountain-valley wind systems develop due to differential heating conditions within
comparatively small distances, which vary between day- and nighttime.

**2.2. How do cyclones form?**
While anti-cyclones are primarily associated with low wind conditions in their center and strong
winds are only found around its edges (i.e. co-located with another pressure system), cyclones
feature typically strong pressure gradients and are thus associated with strong winds and wind
gusts. Many extra-tropical cyclones develop under the influence of the mid-latitude jet-stream,
its associated baroclinicity and upper-air flow divergence. Other cyclones develop as
secondary cyclones in the trailing cold fronts of pre-existing systems and are more influenced
by lower-level processes such as latent heat release (Parker, 1998; Dacre and Gray, 2009).
Another large group of cyclones develop by the interaction of atmospheric waves with
topography (McGinley, 1982; Radinovic, 1986). Focusing on the North Atlantic sector for a
European perspective, baroclinically driven (primary) cyclones develop typically over the North
Atlantic (Dacre and Gray, 2009), secondary cyclones develop further downstream often close
to the eastern North Atlantic (Priestley et al., 2020a), and the orographically driven cyclones
dominate in the Mediterranean basin (Trigo et al., 1999).

The most common conceptual models to describe extra-tropical cyclone development are the
Norwegian and the Shapiro-Keyser models (Bjerknes, 1922; Schultz et al. 2019; Dacre 2020).
According to the Norwegian model, a stationary front forms between cold and warm air,
initiating strong vertical wind shear within the troposphere. A front is a density discontinuity
and, hence, separates cold and warm air masses. Typically triggered by an upper-level trough,
a cyclone begins to grow along this front where it develops a warm and a cold front. As the
cyclone deepens, both fronts become better defined and a warm sector develops. When the
cold front catches up to the warm front, the so-called occlusion process starts. At this stage,
the cyclone reaches its most intense period (Bjerknes 1922), followed by cyclone decay. In
the Shapiro-Keyser model, the initial development is similar, but the cold front does not
overtake the warm front, but rather builds a T-bone structure (see Fig. 16-24 of Schultz et al.,
1999) instead of a narrowing warm sector during occlusion as in the classical model (Shapiro
and Keyser 1990).

Windstorms produce winds which are strong enough to cause damage; they typically have
windspeeds in excess of 15m/s (Wallace and Hobbs 2006). In order to quantify the impact of
windstorms, it is important to know the parts of a storm where the strongest wind speed
typically occurs. There are three zones where strong winds can occur: the warm jet, the cold
jet, and the sting jet (Clark and Gray, 2018). Hewson and Neu (2015; see their Fig. 1) have
developed a conceptual windstorm model to describe how strong winds may develop
associated with the passage of a cyclone during different stages of its development. In most
cases, the strongest winds are often associated with the passage of the cold jet at the cold
front. However, Shapiro-Keyser cyclones may on occasion feature sting jets, which, if they
reach the surface, may lead to even more damaging wind speed (Clark and Gray, 2018).

The potentially most damaging events affecting Europe are commonly assigned to slow
movers, rapid developers, or serial storms (Mailier et al. 2006). Slow mover cyclones lead to
large accumulations of precipitation in the same area, often triggering severe floodings (Grams
et al., 2014). Rapid developers are fast deepening cyclones, often fulfilling the conditions for
a "bomb" (Gyakum and Danielson, 2000). When occurring close to Europe, many of these are
secondary cyclones. Finally, serial storms (also known as cyclone families) indicate that
multiple and related cyclones affect the same area within a comparatively short period of time,
leading, potentially, to severe cumulative losses (Mailier et al., 2006; Pinto et al., 2014). In
these clustering periods, the passage of storms occurs more frequently than may be expected
if they would occur independently from each other (Vitolo et al., 2009; Franzke, 2013; Blender
et al., 2015). Two physical reasons are given in the literature (Economou et al., 2015; Dacre
and Gray 2020): i) the steering through the large-scale flow, typically characterized by an
intensified, quasi-stationary jet-stream extending towards Europe and ii) the occurrence of
secondary cyclogenesis.

**2.3. Spatial characteristics of storms**
To analyse cyclones and storms, objective identification and tracking methods are needed
(Ulbrich et al., 2009; Neu et al., 2013). This leads to a Lagrangian perspective where certain
properties during the life cycle of the cyclone can be defined by e.g. radius, propagation speed,
and spatial wind distribution. Various objective methods for the identification and tracking of
extra-tropical cyclones have been used to investigate their characteristics (Neu et al., 2013;
Priestley et al., 2020a).

In the North Atlantic-European region, cyclone track densities show maximum values over the
western North Atlantic with a second maximum over the Mediterranean (Ulbrich et al., 2009;
Pinto et al., 2005). North Atlantic cyclone activity shows a tilt towards the northern North
Atlantic. While this can be found in different reanalysis products, Coupled Model
Intercomparison Project phase 5 (CMIP5) simulations are characterised by a bias of the
maximum and tilt in the North Atlantic, leading to more zonally oriented storm tracks (Zappa
et al. 2013). While many cyclones can be identified in the extra-tropics, only a subset of strong
cyclones lead to a high wind speed. See section 4.1 for related storm indices.

**2.4. Large**-s**cale circulation characteristics and their impact on wind**
Storms and their related wind gusts are local in nature. Nonetheless, the large-scale
background circulation can still provide information in which areas strong winds are likely to
occur. Here, we apply the concept of atmospheric weather regimes (Hannachi et al., 2017) to
determine the characteristics of the large-scale circulation. Atmospheric weather regimes are
recurrent, dynamically relevant circulation patterns and allow the description of low-frequency
variability due to transitions between distinct regimes. Because of their preferred occurrence
locations, they potentially provide prediction and downscaling possibilities for smaller scale
weather events and extremes (Cassou et al. 2005).

To demonstrate the relation of specific regimes to preferred jet-stream patterns and storms,
we show weather regimes based on sea-level pressure fields from the latest European Centre
for Medium-Range Weather Forecasts Reanalysis (ERA5) (Hersbach et al. 2020) over the
North-Atlantic-Eurasian region (30°N-90°N, 90°W-90°E) for winters (December through
March, DJFM) from 1979-2020. The details of the applied regime analysis are described in
Crasemann et al. (2017). We identify 5 regime states (Fig. 1a): (1) Scandinavian/Ural blocking
(SCA-URAL BL), (2) the North Atlantic oscillation in the positive phase (NAO+), (3) blocking
over the North Atlantic (ATL BL), (4) North Atlantic trough (ATL TR) and (5) the NAO in its
negative phase (NAO-).

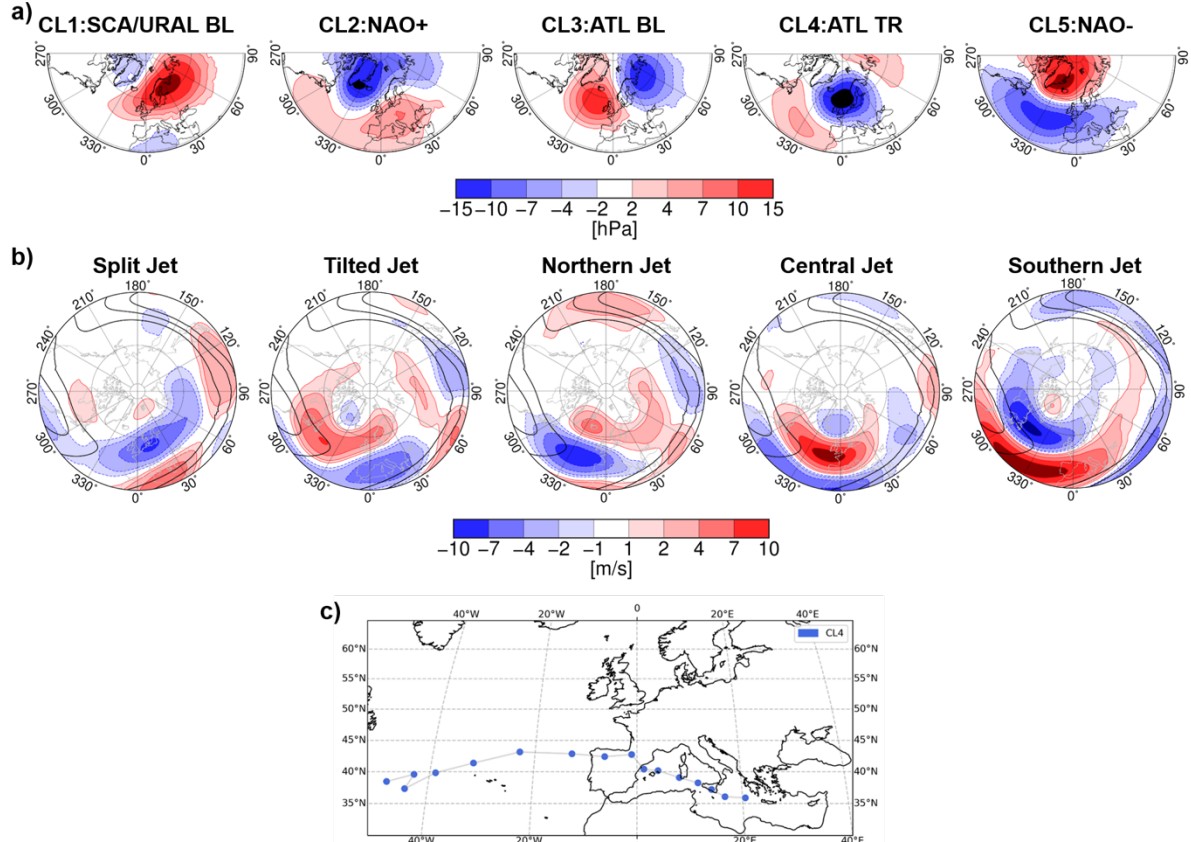

Figure 1: a) Weather regimes determined from ERA5 reanalysis data for December to March
(DJFM). Shown are the regime patterns in terms of sea-level pressure anomalies (shading),
black contours indicate the climatology for DJFM, shown are isolines at 1000, 1005, 1010 and
1015 hPa. b) The jet-stream patterns associated with the individual weather regimes, obtained
by composites of zonal wind anomalies at 250 hPa (shading), black contours indicate the
zonal wind climatology for DJFM, shown are isolines at 20, 30, and 40 m/s. c) eXtreme
WindStorms XWS data base (Roberts et al. 2014) track for storm Klaus based on ERA5 data,
identified with the method of Leckebusch et al. (2008). Storms are defined by the exceedance
of the local 98th percentile of near surface wind speed. Each dot represents the position of
the wind field center of storm Klaus for 6 hour time steps from 22 January 2009, 06:00 to 26
January 2009, 00:00. The color of the dots shows the weather regime of that date.
The characteristic patterns for the jet-stream associated with these weather regimes have
been obtained by compositing the zonal wind anomalies at 250 hPa over the days assigned
to each regime. The five jet-stream patterns (Fig. 1b) are very similar to those obtained by
previous studies (Dorrington and Strommen, 2020; Woollings et al., 2010; Franzke et al.,
295  2011).


The regime analysis assigns the atmospheric circulation of each day over the period 1979-
2020 to one specific cluster and enables a characterization of the large-scale background for
specific windstorm events. As one example, Fig. 1c shows the eastward movement of the
extreme storm Klaus from Jan 22 to Jan 26, 2009 along an unusual southerly path. The storm
'Klaus' was characterized by strong and record-breaking wind speed over northern Iberia and
southern France. During the formation, intensification, and eastward movement of Klaus, the
Atlantic trough weather regime associated with the central jet-stream configuration prevails
(Fig. 1c). This central jet-stream pattern sets the necessary large-scale background flow for
the development and movement of this extreme storm (Liberato et al., 2011).

306

The concept of weather regimes enables the characterization of the large-scale atmospheric
circulation, in particular the jet-stream pattern, during extreme storm events. If changes in the
occurrence of these extremes can be related to an anomalous frequency of occurrence of a
specific weather regime, the use of these regime states offers potential predictability of large-
as well as small-scale wind impacts.

312

**2.5. Temporal characteristics of storms and seasonal variability**
The occurrence of extreme wind speed and storms is subject to a strong seasonal pattern in
Europe. According to Young et al. (1999), windstorms occur 30% more frequently in winter
than in summer (see also Fig. S1). We compared the wind gusts from three reanalysis
products (ERA5 (Hersbach et al., 2020), COSMO-REA6 (Bollmeyer et al., 2015) and COSMO-
REA2 (Wahl et al., 2017)), to 145 German station observations (Kaspar et al., 2013). While a
direct comparison is difficult, qualitative statements on seasonality can be made with all data
sets. The number of occurrences of wind gusts is determined for certain wind speed intervals,
which are shown against the warning levels (WL) of the Deutscher Wetterdienst (DWD). The
warning levels are defined by 6 different wind speed thresholds: 14, 18, 25, 29, 33, and 39
m/s (Primo, 2016), referring to 4 WL (WL1- WL4), with WL2 and WL3 being divided into two
intervals (DWD, 2021). Compared to observations, wind gust frequencies are underestimated
in reanalyses. The higher the wind gusts, the higher the underestimation. Therefore, COSMO-
REA2 shows a significantly better agreement with the reference, especially for WL3 in summer
and WL4 in winter. The benefit of the higher resolution provided by regional reanalyses
compared to their global counterparts is well documented for near surface wind speed
(Niermann et al., 2019). Results shown in Fig. S1 emphasize the importance of using high
resolution models to represent extreme wind gusts in reanalysis products.

331

Above 25 m/s there is a clear difference between summer and winter months, which becomes
stronger the higher wind speed is considered. In summer, wind speed over 30 m/s does not
appear in the coarser reanalysis products ERA5 (~30km) and COSMO-REA6 (~6km) at all
and for the high-resolution reanalysis COSMO-REA2 (2km) and the point observations the
occurrence of wind gusts of WL3 or WL4 in summer is smaller than in winter by a factor of 10
to 100.


The intra-annual variability is not only visible in meteorological data but also in loss data from
insurance companies (Hoeppe 2017, Franzke 2017), which shows the strong impact of storms
and especially winter storms on society and economic areas (Klawa and Ulbrich, 2003). The
energy sector is strongly affected by the occurrence of windstorms, and especially their
seasonal variability. Due to the worldwide effort to convert the energy system to renewable
sources, the industry will have to deal more with seasonal fluctuations in energy availability.
The interest and the need for precise knowledge of the wind conditions in various regions is
therefore growing, as energy production directly depends on it; for more details about wind-
based energy production please see section 4.6.

### 349 2.6. Winds induced by convective activity

Most of the wind damage in temperate latitudes is due to extra-tropical cyclones. However,
damage can also occur to structures, crops and forests from winds produced by convective
storms (Gatzen et al., 2020; Parodi et al., 2019); since our focus is more on extra-tropical
storms, we keep this part rather brief. The following conditions need to be met for convection
to occur (Wallace and Hobbs 2006): (1) The atmosphere needs to be conditionally unstable,
(2) there needs to be a reservoir of substantial moisture in the boundary layer, and (3) there
needs to be sufficient lifting due to low level convergence to cross the threshold to start the
instability.

Convective systems and storms can lead to severe wind speed connected to tornadoes, gust
fronts and downbursts (Wallace and Hobbs 2006). Tornadoes are rapidly rotating air systems
which connect with the ground and can lead to devastatingly strong winds. Downbursts are
downward directed winds due to the negative buoyancy of the downdraft air. Convective
storms can also have gust fronts. The gust fronts form due to downdrafts in the convective
storm forming a pool of cold, dense air which replaces the warmer, buoyant air of the
environment.

These downdrafts can lead to severe wind gust speeds at the surface (Bunkers and Hjelmfelt,
2021) with speeds of up to 42 m/s. So far, relatively little attention has been paid to wind
damage to infrastructure, forests and agriculture from such events besides the studies by Jim
and Liu (1997) and Peterson (2000). Forest damage from thunderstorms in areas, which
previously were rarely affected, such as eastern parts of Europe ( Nosnikau et al., 2018; Sulik
and Kejna, 2020), but have experienced an increase in convectively available potential energy
and near surface moisture which can cause more thunderstorm activity (Taszarek et al. 2021).
It is expected that anthropogenic global warming will lead to an increase of convective storms
(Lepore et al. 2021; Taszarek et al. 2021; Diffenbaugh et al. 2013).

Another type of convective storm is derechos, which are a clustering of downbursts, organized
by a line of thunderstorms (also called a squall line), that lead to widespread straight-line
winds, and can cause damaging winds. They occur frequently in the Great Plains area of the
USA (Ashley and Mote, 2005) but can occur around the world, including Central and Eastern
Europe (Gatzen et al., 2020). Some examples of the devastating impact of derechos on forests
are described in Goff et al. (2021), and Negrón-Juárez et al. (2010).

## 3 Wind-surface interaction

### 3.1. The physics of fine scale interactions between surfaces and wind

The characteristics of the wind speed and gustiness in a given environment are dependent on
surface characteristics, such as its roughness, all of which are highly influential on the levels
of damage caused. The momentum of the mean horizontal wind is vertically transferred by
turbulence, i.e. near the surface, large whirling air packages break up into smaller ones and
their momentum dissipates into thermal energy or is absorbed by roughness elements, such
as trees and buildings. The strength of the wind is altered by topography and the roughness
of the surface (Stull, 2017; Kaimal and Finnigan, 1994; Finnigan et al. 2020). Thus, the
damage level can vary dramatically at small scales (Gromke and Ruck 2018; Forzieri et al.
394 2019).


Typically, the boundary layer above the Earth's surface is subdivided into three sublayers: 1)
a roughness sublayer that is characterized by the flow around obstacles and varies locally and
where mechanical turbulence dominates, 2) one or more inertial sublayers, where the
influence of the individual obstacles and surfaces is blended together and the vertical energy
fluxes are constant with height and 3) a mixing layer above, where the Coriolis force gains
influence and is often separated from the free atmosphere by a capping inversion and an
entrainment zone (Stull, 1988; Kaimal and Finnigan, 1994). The effect of buoyancy and
thermal stability is very important for the formation of strong winds, i.e. for cyclones and
thunderstorms. During storm events, high wind speed increases friction within the lower
boundary layer and also increases form drag by obstacles. The instability of the shear in the
flow created by the drag of the surface leads to turbulence, which affects the vertical exchange
of mass, momentum, and scalars. Thermal gradients near the surface are reduced or
disappear due to this mixing, which results in neutral stratification near the surface, i.e. thermal
stability need not be considered in the equations of the vertical wind profile (Stull, 1988).

As turbulent movements play a major role in the momentum transfer to the surface it is
important to regard shear forces and gustiness as the damaging characteristics of the wind
field (Gromke and Ruck 2018). For example, in forest ecosystems trees are blown down at a
mean wind speed considerably lower than those estimated by pulling experiments (Oliver and
Mayhead, 1974; Milne, 1991). Boundary-layer eddies create a local increase in wind speed
and windshear close to the surface (Romanic und Hangan, 2020) and leading to coherent
eddies (Raupach et al., 1996). The loading due to these turbulent structures with higher energy
and momentum can be accounted for in a gust factor (Hale et al., 2015; Chen et al., 2018;
Holland et al., 2006; Usbeck et al., 2010). Since trees react to gusts like damped harmonic
oscillators (Mayer, 1987; Gardiner, 1992) there has been considerable debate about whether
the arrival frequencies of these coherent eddies could lead to resonant failure (Gardiner, 1995;
Peltola, 1996); however, this does not happen (Schindler and Mohr, 2019; Schindler and
Kolbe, 2020; Kamimura et al., 2022), probably due to the efficient damping of trees (Spatz and
Theckes, 2013). Besides the drag force of a plant (Rudnicki et al., 2004; Queck et al., 2012;
Vollsinger et al. 2005), the level of damage depends also on the acclimation of plants to the
wind (Telewski, 1995; Nicoll et al., 2019), which is a function of the maximum wind speed
(Bonnesoeur et al., 2016; Dèfossez et al., 2022). They are adapted to wind forces and build
stronger roots and wood structures depending on the main wind direction and magnitude
(Nicoll and Ray 1996; Tomczak et al. 2020).

Furthermore, the development of turbulence above and within the canopy is different between
naturally uneven aged woods and managed forests or plantations. Experiments showed that
the inflection of the wind profile (i.e. maximum gradient of wind speed) is weaker in
heterogeneous compared to homogeneous canopies, and that it occurs deeper within the
canopy, i.e. the displacement height is lower (Cionco, 1972; Belcher et al., 2012; Queck et al.
2016). Furthermore, homogeneous forests are more vulnerable than naturally uneven aged
woods (Everham and Brokaw, 1996; Mitchell 2013). Obviously, the adaptation to wind stress
is not restricted to single trees but extends to the structure of natural mixed woods too. The
characteristics of the tree (height, diameter, canopy size, wood properties), and the tree
resistance to uprooting and breakage are all affected by the level of wind exposure (Gardiner
et al. 2016). Recent experimental measurements of tree damage during a super typhoon
(Kamimura et al. (2022) has also shown that collisions between the crowns of individual trees
and the crowns of their neighbours is extremely important in reducing tree movement during
strong winds and contributing to their overall stability. These adaptations of plants to living in
a windy environment must be considered when modelling the risk of wind damage to tree
stands.

Large eddy simulations (LES) are used to better understand the complex current patterns and
the acting wind forces near heterogeneous surfaces (Stoll et al. 2016, Takemi et al. 2020).
These turbulence-resolving models include all the basic physical equations; however, they
require considerable computer resources and are therefore unsuitable for operational use.
Simplified mechanistic models (Holland et al. 2006, Gross et al. 2018, Duperat et al. 2021)
parameterize the turbulence spectrum and operate on a larger spatial scale; thus, need less
computational resources. Statistical approaches (Jung and Schindler 2015, Dupont 2016)
focus on predicting critical thresholds at which wind damage occurs and are therefore efficient
for operational damage prediction. The indices discussed in section 4 are based on empirical
observations and have proven useful in a wide range of applications.

**3.2. Mean wind and gust rates for different landscapes**
The gustiness of the wind is critically important for assessing the likely impact of strong winds
on forests, agriculture, and structures (Usbeck et al. 2010; Gardiner et al. 2016). The level of
gustiness is known to be influenced by surface roughness (Table 1), the height above the
ground, and wind speed (Ashcroft 1994; Verkaik 2000). Gust ratios are also affected by wind
speed (see Born et al., 2012; their Fig. 2) and by the type of storm (Krayer and Marshall 1992;
Harper et al. 2010).

| Roughness Class | Aerodynamic roughness length (m) | Gust Ratio (3 s to 10 min) | Gust Ratio (3 s to 60 min) |
|---|---|---|---|
| **1** | 0.003 | 1.36 | 1.44 |

| | | | |
|---|---|---|---|
| **2** | 0.01 | 1.42 | 1.49 |
| **3** | 0.03 | 1.48 | 1.56 |
| **4** | 0.1 | 1.58 | 1.66 |
| **5** | 0.3 | 1.74 | 1.85 |

Table 1. Wind rate (mean/gusts) for different landscapes. 3 s gust to 10 min and 60 min mean wind at 10 m height, by terrain category. From Ashcroft (1994). Roughness Classes: 1: off-sea wind onto flat coastal areas; 2: level grass plains, e.g. marsh; 3: standard category: fairly level terrain-mostly open fields with a few houses and buildings; 4: fairly level terrain with more hedges, trees and villages, farm buildings; 5: many trees and hedges, or fairly level wooded country or more open suburban areas.

## 4.      Wind and storm related indices and critical thresholds

Wind and storm related indices and thresholds are a vital tool in assessing the likelihood and magnitude of damage. While there are many definitions for indices and thresholds, here we define an index as a number or a category, serving as an aggregated measure of a quality, which can be reached by means of observation, arithmetic calculation, or different modelling techniques. A threshold is defined here as a value taken or calculated from a numerical or a categorical range, and when the threshold value is crossed, it indicates a significant increase in the probability for an event to take place or for a certain condition to be fulfilled. Indices can be used to predict damage caused directly by wind or a storm, or when wind modulates the damage caused by another process such as fire or drought. Since indices and thresholds can be as effective as complex mechanistic models but more cost-effective, it is of no surprise that there is a plethora of indices. There are general indices that are not bound to a given sector or environment, but many of the indices and thresholds available require a careful selection process according to the target ecosystem. Below we provide an extensive review of available indices, focusing on five key terrestrial sectors.

### 4.1. General storm indices: scale and severity indices

Classical wind scales are defined by phenomena caused by the interactions between wind and the surface. A very prominent example is given by the Beaufort scale (Stull, 2017). It classifies the effect of wind on wave generation, tree movement and the damage of buildings. Similar scales exist for tornados, e.g., the Fujita scale and the Torro scale (Kirk, 2014), which relates the tornado intensity to damage description. As short gusts and shear forces are very important factors of storm risk, the Enhanced Fujita scale includes further information on derived maximal tangential 3s gust speeds (Fujita, 1981). Recently an improved wind speed scale and damage description has been suggested for Central Europe (Feuerstein et al., 2011). Finally, The Saffir–Simpson hurricane wind scale (Ellis et al., 2020) is based on the highest wind speed averaged over a one-minute interval 10 m above the surface. It can provide some indication of the potential damage a hurricane will cause upon landfall.

Several storm severity indices have been developed to quantify the severity of a windstorm regardless of the land cover. These indices are used to identify severe winter storms and analyze their impacts and to investigate storm trends in past and future climate conditions. They often include the cube of the wind speed, assuming a proportionality of the dissipation

rate of the wind kinetic energy to damage. A selection of these indices is presented in Table
S1.

From an historical context, one of the earliest storm severity indices was developed by Lamb
(1991) to grade and rank storms based on the greatest observed wind speed over land, the
area affected by damaging winds and the overall duration of occurrence of damaging winds.
Later, in a study by Klawa and Ulbrich (2003), the wind speed values were scaled with the
local 98th percentile. Based on this approach, Leckebusch et al. (2008) identified and tracked
windstorms in time and space and computed an event-based storm severity index that
quantifies the potential impact of a storm. This index considers the relation of the maximum
daily wind speed to a certain local percentile of maximum daily wind speed (e.g. the 95th or
98th) as well as the affected area. For example, in their study they found a trend for an
increase in severity of storms during 1960–2000 and for 2070-2100 under anthropogenic
climate change conditions. Pinto et al. (2012) extended this approach by taking into account
the exposure and including local population levels in a Loss Index, resulting in the finding that
the maximum storm losses for current climate conditions are likely to be exceeded in the
future. Additionally, Haylock (2011) used a storm severity index to identify the severest storms
for 72 hours storm footprints. This index considers the latitude and the excess of the maximum
wind speed over a 72 hour period taken from six-hourly values over a threshold (e.g. the local
90th percentile of wind speed).

## 4.2. Forests

### 4.2.1. Topographic indices

Many topographic indices have been used for assessing the risk of wind damage to forests
(see Table S2). These indices can be based on elevation, slope characteristics such as
compass angle, aspect, and curvature, or are more complex such as TOPEX (Topographic
exposure; Quine and White, 1998) which was developed as part of a risk assessment method
(Windthrow Hazard Classification) to predict the height at which trees could be expected to be
first damaged (Miller, 1986). TOPEX is the sum of the angle to the horizon in the eight principal
points of the compass and can be calculated for different distances from the point of interest.
Furthermore, such indices can be used to create even more complex predictive systems. For
instance, when TOPEX is combined with elevation and aspect it produces a system called
DAMS (Detailed Aspect Method of Scoring; Quine and White, 1993) for predicting wind speed
variation in the landscape. This system is entirely based on topographic measures and
compares favorably with modelling systems based on solutions of the fluid equations (Suárez
et al., 1999).

The actual variation of wind speed with height above the ground is a function of the surface
roughness and the topography. Predicting variations of wind speed across flat surfaces is
relatively straight forward, especially for strong winds by using a measure of the aerodynamic
roughness of the surface and a logarithmic wind profile (Garratt, 1980; Stull, 1988). Even in
stable or unstable conditions the profile can be modified with the addition of the diabatic term
$\psi_m$ (Kaimal and Finnigan, 1994; Panofsky and Dutton, 1984). Often the roughness of the
surface is simplified into different roughness classes (Troen and Petersen, 1989) to allow for
easier estimation of the surface roughness. However, when even-strong-winds flow over
topography the simple logarithmic profile breaks down and the shape of the wind profile
strongly varies between the upwind slope, the crest of the hill and the downwind slope, where
the flow may even separate (Belcher et al., 2012). Thus, one should not only calculate
topographic indices for the target locations but calculate also for the neighboring areas and
assess the change in value between the target location and its surroundings (Ruel et al., 1997;
Schindler et al., 2012; Murshed and Reed, 2016).

We reviewed the literature, focusing on studies using topographic indices to assess and
predict damage caused by strong winds, as topographic indices are a common feature in
modelling wind damage in forests (Table S3). The most commonly used variables were (Fig.
2): elevation, slope, aspect and TOPEX. We assessed the usefulness of the four most used
topographic indices in modelling forest damage according to their inclusion in final models and
according to the importance/influence metrics reported. We note that most studies employed
a multivariate modelling approach, thus, a certain variable may appear less useful due to
overlap in the variance explained with another variable, but not necessarily due to the
variable's lack of explanatory power (Scott and Mitchell, 2005). Furthermore, there are other
topographic indices that were not tested so far for their contribution in forest damage prediction
(see Florinsky, 2017).

Elevation was useful in about a third of the studies, and was particularly useful when the study
area was very large, encompassing an entire region, state or country (Díaz-Yáñez et al., 2019;
Kramer et al., 2001; Torun and Altunel, 2020; Mayer et al., 2005) or when there was a strong
gradient of elevation, preferably reaching above 900 m above sea level (Krejci et al., 2018;
Pasztor et al., 2015; Torun and Altunel, 2020; Kramer et al., 2001; Mayer et al., 2005). The
trend in the correlation between elevation and forest damage was found to be inconclusive, to
be both positive (Díaz-Yáñez et al., 2019; Krejci et al., 2018; Pasztor et al., 2015) and negative
(Mayer et al., 2005; Albrecht et al., 2013), or only present for a certain range of elevation
(Albrecht et al., 2013; Torun and Altunel, 2020;). While there is an expectation for an increase
in forest damage with higher elevation due to an increase in wind speed (Machar et al., 2014),
diversity of trends can stem from the involvement of other topographic indices that may contain
similar information (e.g. slope or TOPEX), and also due to varying levels of acclimation of
trees to the wind conditions present at different elevations (Gardiner 2021).

The slope was shown to be useful in about half of all articles, however it is difficult to observe
a clear relation to forest damage. In articles that identified a contribution of slope, the relation
of damage with slope was found to be either positive (Díaz-Yáñez et al., 2019) or negative
(Mayer et al., 2005; Morimoto et al., 2019; Schütz et al., 2006). But an important deciding
factor can be the aspect of the slope (useful in about 40% of all articles) as there is often an
interaction between the two (Suvanto et al., 2018, 2016; Díaz-Yáñez et al., 2019; Hanewinkel
et al., 2014). In this sense, the aspect likely indicates the forest's susceptibility to wind coming
from a certain direction, as in most cases of usefulness of aspect, the slope was also useful.
Finally, TOPEX was found by 77% of articles as useful, and when a trend was reported, all
studies reported higher damage or probabilities for forest damage being associated with more
exposed locations (Albrecht et al., 2013, 2012; Jung et al., 2016; Morimoto et al., 2019;
Mitchell et al., 2001; Taylor et al., 2019). One of the reasons for TOPEX's usefulness is that it
does not strongly overlap with the information contained in other wind-based variables
(Albrecht et al., 2019; Schindler et al., 2012). However, when TOPEX is calculated only for a
certain cardinal direction (e.g. west) it contains information that is very similar to aspect.

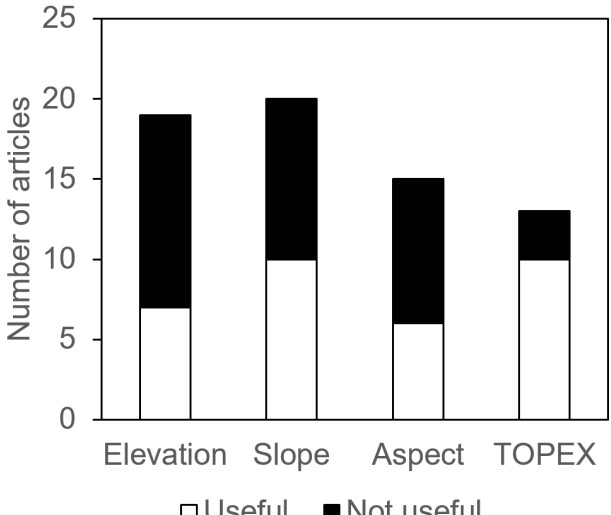


Figure 2. An assessment of the usefulness of the most commonly used topographic indices.

**4.2.2. Fine scale wind and surface interactions**
Interactions between the surface and the wind field are controlled by surface roughness,
absolute wind velocity and atmospheric stability. The commonly used index related to
surface interactions that is relevant for wind and storm damage is the critical wind speed
(CWS). The CWS defines the threshold wind speed for overcoming the maximum resistance
to stem breakage or uprooting of a tree (Gardiner et al., 2016; Peterson et al., 2019; Hale et
al., 2015; Chen et al., 2018; Holland et al., 2006). CWS is a standard term in forest ecology.
The typical averaging interval for CWS is a period of a few minutes, e.g. 3 minutes (Peltola
and Kellomäki, 1993), 10 minutes (Dupont et al., 2015, Peltola et al. 1999) or 60 minutes
(Hale et al., 2015). CWS is estimated either at a height of 10 m above the canopy or at the
tree top at the stand edge.
One of the governing quantities to describe the interactions between wind forces and stem
breakage or uprooting is the applied maximum bending moment (BMmax) (Quine et al.
2021), which is the sum of wind forces in the tree crown and the additional turning moment
due to stem bending and deflection of the stem and crown of a tree (Peltola, 2006). BMmax
calculation refers typically to the mean bending moment (BMmean) and a gust factor (see
e.g., Gardiner et al. 1997). A tree uproots if its BMmax at the ground level exceeds the
resistance of the root–soil plate, and a tree breaks if its BMmax at breast height (1.3 m)
exceeds the critical value of the stem's modulus of rupture (Peltola et al. 1999, Quine et al.,
2020). The gust factor is parameterised by wind measurements (field or wind tunnel) and
depends on the spacing/height ratio of tree stands and the location relative to the
forest/stand edge (Gardiner et al. 1997; Quine et al., 2020).  The wind measurements are
taken from the top of the canopy, and the bending moment is typically determined from the
level of zero-plane displacement (e.g., 0.8 of the tree height; Gardiner et al. (1997)).
Nevertheless, measurements of the effects (Gardiner et al. 1997) as well as directly solved
finite element models of the crown architecture (Ruy et al., 2022) have shown the influence
of crown architecture on the maximum bending moment. Therefore, the gust factor used in
the calculation of BMmax may need to be varied according to stand composition and tree
type.

The probability of occurrence of CWS, as a measure of storm damage risk for specific forest
stands, depends on the statistics of wind velocity, e.g., on hourly maximum synoptic winds
(umax: Usbeck et al., 2010, Chen et al., 2018) or maximum geostrophic wind speed
(Blennow and Olofsson, 2008). CWS is used to parameterize impact models for the
estimation of storm risk in forests such as ORCHIDEE-CAN (Chen et al., 2018),
SWAN/ADCIRC (Akbar et al., 2017), GALES and HWIND (Peltola et al., 1999; Gardiner et
al., 2000, 2008).

The key parameter in the calculation of CWS is the diameter at breast height (DBH), which is
a standard parameter in forest inventories. DBH is commonly defined as the stem diameter at
1.3 m above the ground (Peterson et al., 2019; Gardiner, 2021; Hale et al.; 2015, Chen et al.,
2018; Holland et al., 2006; Hanewinkel et al., 2014; Beck and Dotzek, 2010; Gardiner et al.,
2008; Peltola, 2006). DBH is the most used structural parameter due to its easy and
practicable measurement and due to its widespread application in forest management (Liu et
al., 2018). DBH is also used to derive other structural parameters like tree height and Leaf
Area Index (LAI) which can also be derived from normalized difference vegetation index
(NDVI) as a standard product of satellite remote sensing. These structural quantities are
important both for statistical analysis and for the parameterization of storm risk models.

Other important parameters for calculating CWS are the mean drag coefficient (cd) which is
part of the equation of the drag force (Vogel, 1989; Akbar et al., 2017; Dupont et al., 2015),
turbulence intensity, gust duration(Hale et al., 2015; Chen et al., 2018), tree density (Peterson
et al., 2019; Albrecht et al., 2015), tree height, crown projection area and crown volume
(Peterson et al., 2019; Gardiner, 2010, 2021; Albrecht et al., 2015; Hale et al., 2015; Chen et
al., 2018; Dupont et al. 2015; Peltola, 2006), and tree species (Hanewinkel et al., 2014).
Additionally, the edge factor index describes the influence of a tree's position relative to a
forest edge, the shape of the forest edge and the width of any upwind gap (Chen et al., 2018;
Gardiner et al., 2010; Peltola, 2006).

The severity of storm damage depends on the ability of a tree to resist the applied bending
moment from the wind and on the stability of the root soil complex (Nicoll et al., 2006;). If soil
water content is close to saturation the critical resistive moment of trees (BMcrit) can be
reduced significantly during storm events, which could become increasingly important with the
increasing frequency of heavy winter rain in temperate forests in the context of regional climate
change (Défossez et al., 2021).

The uncertainty of CWS results from the consecutive solving of analytic equations including
accumulated uncertainties of the different input quantities. Additional uncertainties result from
the differences in the models used. Sensitivity tests using GALES (Locatelli et al., 2017) and
HWIND with a variation of the input parameters of +/-20% lead to a more than 20% change in
CWS. For example, CWS is especially sensitive to changes in DBH. The measurement
uncertainty of the DBH ranges between 2 and 10% depending on the absolute diameter (Qin
et al., 2019). Applied in HWIND and GALES the variation of DBH of +/- 20% lead to changes
of CWS of +30% and -46% (Gardiner et al., 2000). The most comprehensive analysis of wind
risk model uncertainty was made by Locatelli et al. (2017) who found that tree DBH, tree height
and inter tree spacing were the most critical factors.

**4.3. Urban areas**
**4.3.1 The urban boundary layer**
The small-scale interactions of the wind field with urban surfaces are significantly different
from natural surfaces due to high three-dimensional variability of impermeable artificial
obstacles (buildings). These differences lead to a higher mean surface roughness of the urban
surface (Grimmond and Oke, 1999; Oke et al., 2017) combined with a general attenuation of
the mean wind speed, the wind speed averaged over some time period (Chen et al., 2020),
as compared with more natural surfaces. The level of increase in roughness depends on the
morphology - density, size, and composition - of the obstacles along the flow direction. The
height of the roughness layer is 2-3 times the mean height of the buildings. Within this layer,
mechanical turbulence generation dominates, and average wind profiles can only be assumed
above the roughness layer, within the inertial sublayer. The averaged roughness of an urban
surface is described by roughness length z0 within equations for vertical wind profiles. This
parameter serves as a useful index for the prediction of turbulent impulse transfer and for
damage prediction, which are derived based on building height, areal fraction and frontal area
index (Grimmond and Oke, 1999). At finer scales, wind speed shows high spatio-temporal
variability. Thus, when using indices based on averaged wind speed, it is also important to
consider that due to the small-scale aerodynamic and thermal heterogeneities of urban
infrastructure (buildings and trees), the local magnitude of the wind speed is temporarily larger
than under rural conditions (Droste et al., 2018). The reasons for this anomaly are again the
inflexibility and impermeability of technical structures and buildings. These features cause
canalization of flows and stronger turbulence generation compared to natural surfaces. There
is also a diurnal-nocturnal distinction in the formation of local thermal wind systems, with street
canyon wind during the day and a nocturnal inflow to the urban heat island (Droste et al., 2018;
Lindén and Holmer, 2011). Thus, indices in urban areas should account for both spatial and
temporal heterogeneities.

### 4.3.2 Indices for estimating damage to individual buildings

Damage occurs either directly by wind pressure or indirectly by the impact at high speed of objects and debris moved by the wind (Tamura, 2009). At the level of individual buildings, air movement results in wind pressure on the building surface and an applied force. Damage to buildings caused by extreme wind loads include resonance and vibration induced damage, damage to roof tiles or sheet roofing, roof lift off and the collapse of walls or entire houses.

The occurrence and type of damage depend on the level of exposure as well as the structural vulnerability of the individual buildings to severe local winds. The European wind loading code EN 1991-1-4 regulates how to adapt the structural design of buildings to the local wind climate. The code defines basic wind velocities for different geographical wind zones based on the 50-year return level of 10 min wind speed at a 10 m height. In Germany, for example, the basic wind velocities range from 22.5 m/s in wind zone 1 (inland areas in southern Germany) up to 30 m/s in wind zone 4 (coastal areas). The basic wind velocities are further adjusted based on the height above ground and the terrain roughness to account for short term wind fluctuations. Terrain roughness is classified in five categories ranging from coastal areas to cities with a high building density. Additionally, where topography (e.g. hills, cliffs etc.) increases wind velocities by more than 5% the effect is taken into account using a topographic index, as the ratio of the mean wind velocity at the height above the terrain to the mean wind velocity above flat terrain. Finally, the wind speed is used to compute the local peak velocity pressure which is a fundamental index for the determination of all wind loads for a specific building (Schmidt 2019). Nonetheless, assigning critical wind speed thresholds to building damage is rather difficult given the heterogeneity of buildings, topography and land-cover.

### 4.3.3 Storm loss models: estimating damage on a district level

Often there is little to no information on the actual damage to individual buildings or small-scale urban structures. Instead, storm loss models come into play, and they relate wind speed to actual building damage data, usually by applying statistical modeling techniques. In some cases, these models rely on the use of wind indices like the exceedance of local wind speed over a critical threshold to calculate monetary loss. In other cases, the model itself calculates a damage index. The purpose of storm loss models is, among other things, to assess current risk to residential structures or to estimate expected losses in future climate conditions. It is often assumed that the maximum daily gust speed (24-hour maximum) is the most influential factor compared to other wind parameters like daily mean wind speed or wind direction and is commonly used in indices as well as in loss models (Donat et al., 2011; Klawa & Ulbrich, 2003; Koks & Haer 2020; Leckebusch et al., 2008; Pardowitz et al., 2016; Welker et al., 2021).

Building damage data on a district level is usually provided by insurance companies and is analyzed in the form of the loss ratio, which is the amount of insured loss per day and district, divided by the corresponding sum of insured value, or claim ratio, which is the number of

affected insurance contracts per day and district, divided by the corresponding total number
of insurance contracts (Prahl et al. 2015).

The functional relationships between wind and damage are usually referred to as damage
functions. As the relationship between damage and wind depends strongly on local conditions
like building or city structure, there is no universal function or model and instead a variety of
different damage function formulations are in use. A detailed overview can be found in Prahl
et al. (2015). Power-law damage functions are common. Different exponents for these
functions can be found in the literature ranging from 2 to 12 (Münchener Rückversicherungs-
Gesellschaft. 1993; Heneka et al 2006; Prahl et al. 2012). Some damage functions also
assume an exponential form (Prahl et al., 2015).

Another type of model are probabilistic models which calculate the probability that a certain
loss threshold is exceeded (Pardowitz et al., 2016; Prahl et al., 2012). Some examples of
existing models are shown in Table 2. Most models still need to be fitted to local conditions
and validated with existing damage data. Model selection depends on the available data.

| | |
|---|---|
| $L(v_{max}) = 2.48 * 10^7 * exp(0.48v)$ | Dorland et al. (1999) |
| $D(v) = \left(\dfrac{v_{max}}{v_{98}} - 1\right)^3$ | Klawa & Ulbrich (2003) |
| $LR(v, f(v_{crit}), \varDelta v) = \displaystyle\int_{-\infty}^{v} f(v_{crit})G(v)dv_{crit}$ $G(v) = \begin{cases} 0, v < v_{crit} \\ D(v), v_{crit} \leq v \leq v_{tot} \\ 1, v_{tot} \leq v \end{cases}$ $D(v) = \left(\dfrac{v - v_{crit}}{v_{tot} - v_{crit}}\right)^2$ | Heneka et al. (2006) |
| $P(LR > th) = \dfrac{exp(a + b * v)}{1 + exp(a + b * v)}$ | Pardowitz et al. (2016) |

Table 2: A selection of damage functions including exponential damage relationships (Dorland et al. 1999), power law damage functions (Klawa & Ulbrich 2003, Heneka et al. 2006) and probabilistic damage functions (Pardowitz et al. 2016). *a*, *b* denote coefficients, *D* a damage index, $f(v_{crit})$ a normal distribution of the critical wind speed, *G* a damage ratio, *L* a loss, *LR* a loss ratio, *P(LR>th)* a probability that a certain loss threshold will be exceeded, *th* a loss threshold, *v* a mean daily wind speed, $v_{98}$ the 98$^{th}$ percentile of the local wind speed, $v_{crit}$ a critical wind speed at which buildings are assumed to suffer damage (comparable to the CWS used for trees), $v_{max}$ the maximum daily gust speed, and $v_{tot}$ the buildings total wind speed at which maximum damage is reached.

## 4.4. Transport

Transport systems are the backbones of modern societies. Disruptions within the transport systems can have serious cascading effects that can cause large costs. Weather in general, and windstorms in particular, can affect all aspects and functions of transport systems (Leviäkangas et al. 2011). However, relevant thresholds of wind speed and their impacts are different depending on the mode of transport. Vajda et al. (2014) identify three wind gust thresholds of increasing magnitude, which they relate to general impacts and consequences within different parts of the European transport system: (i) Wind gusts >17 m/s: Adverse impacts on the transport system may start to occur, especially if the resilience of the exposed part of the system is low, but disruptions are rather local. For example, some windthrow of trees can occur along railways and roads, leading to local problems with road and rail traffic. Furthermore, operation of smaller boats could be suspended due to reduced maneuverability, (ii) Wind gusts > 25 m/s: Some adverse impacts can be expected, such as windthrow and electricity cuts occurring on a larger scale. In addition, delays and cancellations in air, rail, road traffic and disturbances of ferry traffic can be expected, and (iii) Wind gusts > 32 m/s adverse impacts are very likely to occur, windthrow of trees can be expected on a large scale, leading to long lasting power failures and delays, and cancellation of rail and road traffic. Furthermore, damage to traffic control devices and structures can occur, airports can be closed, and ferries stay in harbour due to reduced visibility and high waves.

792

The effect of wind on road safety is not extensively explored in the literature (Theofilatos and Yannis 2014). In general, the number of road vehicle crashes caused by strong wind is small compared to the total number of crashes (Edwards, 1998). However, studies have identified specific types of crashes which typically occur under strong wind conditions: overturning, side slip and rotation crashes (Baker, 1986), with trucks, vans, or buses being particularly affected (Becker et al. 2022, Baker 1992). A critical rollover wind velocity of 20 m/s was found for high-sided lorries in crosswind situations (Snaebjornsson et al. 2007). Particularly dangerous situations with strong crosswinds can occur on bridges (Wang et al., 2014; Charuvisit et al., 2004). A vehicle overturning model is applied by the British Meteorological Office (Hemingway et al. 2020). It estimates the risk of overturning based on wind gust thresholds ranging from 23 to 45 m/s, depending on vehicle type, loading, driving speed and wind direction. In addition to direct effects of high wind speed on road vehicles, indirect effects like blocked roads due to falling trees or drifting snow can affect road transport (Leviäkangas et al., 2011).

806

The most frequent impact of high wind speed on railway transport is the blockage of tracks due to windthrow of trees or drifting snow, as well as loss of electricity due to damaged overhead lines (Leviäkangas et al. 2011), an example of a compound event. Only in rare cases, extreme gusts exceeding 40 m/s can blow trains off the track (Sprenger et al., 2017). Mean winds above 17 m/s or wind gusts above 30 m/s have been identified as thresholds relevant for wind induced damage to railway transport (Thornes and Davis 2002). Shaking of overhead cables can cause damage to masts and pantographs on trains. Consequences of windthrow can be collisions of trains with fallen trees. Precursory measures to prevent collisions are reduced traveling speeds or cancelling/limiting train services, commonly leading to widespread delays.

817

The most common impacts on ports are delays due to the disruption of loading and unloading procedures, as well as direct damage to infrastructure. For example, maximum wind speed recommended for crane operations are around 18 m/s, depending on the design of the crane (TT Club et al. 2011). This can have effects on the overall efficiency of ports (Garcia-Alonso et al., 2020). From 88 disruptive events affecting ports and their surrounding seas in the UK between 1950 and 2014, 36% were attributed to wind storms and 12% to storm surges, while the others were mainly related to human error and mechanical faults (Adam et al. 2016).

825

In the case of inland waterway transport, there is generally no large impact of wind on vessels, since they are sufficiently wide and stable (Leviäkangas et al. 2011). However, at specific locations with high local wind speed due to topography or at locations which are difficult to navigate, navigation of pushed convoys without bow thrusters may be suspended in case of high wind speed. In addition to location-specific issues, the vulnerability of vessels to strong wind is strongly dependent on the vessel's characteristics (Schweighofer, 2014). For specific types of inland container vessel mean wind speed of 18 m/s can lead to flooding of open cargo-holds due to heeling and rolling (Hofman and Bačkalov 2010) and increase the risk of sliding of empty containers on the upper tiers.


In the case of deep-sea shipping, vessels like large container ships are rarely lost at sea.
However, high wind speed impose the danger of container losses (Allianz, 2019). The global
average annual loss of containers is estimated to be up to 10,000 per year (Frey and
DeVogelaere 2014). These numbers are low compared to a total number of more than 200
million containers transported per year, but each container lost at sea can lead to a significant
safety and environmental hazard. In contrast to container ships, losses of dry bulk carriers are
often related to heavy weather conditions (INTERCARGO, 2018). Forecasts of ocean surface
conditions are important for route planning to avoid areas affected by windstorms (Kite-Powell
844 2011).


Airplanes are affected by strong winds mainly during take-off and landing. Dangerous
situations related to wind are mainly caused by abrupt changes in wind speed due to wind
gusts, wind shear or microbursts (strong downward movements of air within and below
thunderstorms). In the USA, for example, 48% of weather-related aviation accidents are due
to adverse wind conditions, and of those wind-related accidents 34% are due to crosswinds
and 29% due to wind gusts (Jenama and Kumar, 2013). Therefore, for safety reasons,
separation distances between airplanes are increased under high-wind conditions.
Furthermore, depending on the wind direction, runways may need to be closed. At London
Heathrow, for example, tailwinds of more than 2.6 m/s and crosswinds above 13 m/s are
avoided by changing flight direction or runways (Pejovic et al. 2009). This can lead to delays,
diversions and cancellations of flights.  At London Heathrow Airport, an increase in wind speed
of 0.5 m/s above the mean increases the probability of delay by 8% (Pejovic et al. 2009).

**4.5. Agriculture**
**4.5.1 Wind damage in the agriculture**
Agricultural production levels are crucial for the worldwide economy. Wind leads to substantial
environmental, social, and economic losses and has distinct impacts on agriculture: physical
damage to crops and related infrastructure, soil erosion including nutrient and soil carbon
removal, dust storms, higher evapotranspiration rates of plants, as well as negative impacts
on flowering, pollinators and fruits (Torshizi et al. 2020).

Wind can damage crops through various mechanisms. Most vegetables already react to low
wind speed of around 4 m/s with physiological adaptations that affect the quantity or quality of
the harvest (Rouse and Hodges, 2004). Most kinds of crops can also be directly damaged by
abrasion from windblown dust particles or rubbing leaves (Brandle et al., 2004). In orchards,
wind can cause a considerable loss by breaking branches or damaging the fruit set (Gardiner
et al., 2016). For cereals, lodging (i.e. flattening) is probably the most important impact of wind
(Berry et al. 2004). For instance, wheat yield is usually reduced about 25% when fields are
lodged (Baker et al., 2014), but the loss can reach up to 50-68% (Berry and Spink, 2012) and
the yield of other cereals can decrease by 35-50% under these conditions (Rajkumara, 2008).
In most cases lodging is caused by strong wind accompanied by heavy rain, whereby the
maximum wind speed is the critical parameter (Mohammadi et al., 2020; Niu et al., 2016). The
vulnerability of plants to lodging depends on many factors, for example, excessive usage of
nitrogen fertilizers increases lodging vulnerability of wheat (Berry et al., 2019). It is therefore
difficult to determine general threshold values for a critical wind speed. However, typical
lodging threshold wind speeds at 10 m above the ground for maize, oilseed rape, oats and
wheat can be assumed to be 11.5, 14.8, 15.1 and 16.5 m/s respectively (Joseph et al., 2020;
Baker et al., 2014).

In general, plants exposed to wind are shorter and have thicker leaves and mature plants are
less vulnerable to wind stress than younger plants (Brandle et al., 2004). Therefore, land users
must carefully balance between the investment in wind adaptation measures and yields
(Wiréhn et al. 2020). However, the careful selection of wind resistant varieties with short stems
(Berry et al., 2014), climate resilient plants, or the use of cultivar mixtures can significantly
improve wind lodging stress resistance, as demonstrated in wheat (Kong et al. 2022). Field
fruits react differently to wind exposure: vegetables in general have a very low tolerance to
wind stress, cucumber, pepper, and cabbage for example can be damaged by even a low
wind speed of around 5 m/s, corn and cotton are a bit more resistant than most vegetables,
but also susceptible to wind damage when wind speed exceeds 6 m/s (Rouse & Hodges
2004). Overall, critical thresholds for damage linked to wind speed varies substantially.

Whether or not wind-related agricultural damage will increase under continued warming is
unclear. Peña-Angulo et al. (2020) found that none of the five metrics linked to wind speed
show a significant trend in either direction. However, the results are subject to considerable
uncertainty given that convective events, which are associated with downbursts and straight-
line winds, are poorly simulated in the current generation of global circulation models.

**4.5.2 Wind erosion, dust storms and agricultural drought**
In regions with open and sandy arable land, wind can cause wind erosion and dust storms.
Wind erosion refers to the loss of fertile topsoil, whereas dust storms are singular events where
strong winds displace huge amounts of soil in a short time. Dust storms are particularly
frequent in the so-called dust belt reaching from the north of Africa through the Middle east to
central Asia (Gholizadeh et al., 2021). However, soil loss due to wind erosion is also an
important issue in less erosion-prone areas such as Europe (Borrelli et al., 2017). While wind
is the main forcing factor, there are other climatic factors such as precipitation, soil moisture
and radiation which affects the soil surface and thus influence soil erosion (Bärring et al. 2003).

The threshold values for the mean wind speed at which soil particles start to be dislodged vary
greatly depending on the type and condition of the soil (Shahabinejad et al., 2019). According
to Rouse and Hodges (2004) the minimum mean wind speed to create erosion is normally
about 5-6 m/s at 30 cm above the ground. Shahabinejad et al. (2019) found CWS values of
5.7-8.9 m/s at 10 m height for soils in Iran. Plants can suffer from dust storms due to loss of
plant tissue through abrasion resulting in reduced photosynthesis and burial of seedlings
(Stefanski and Sivakumar, 2009). This can result in considerable economic losses for farmers.
For example, Gholizadeh et al. (2021) demonstrate that a dust storm lasting one hour can
reduce the annual income of farmers by up to 1.2%. Erosion reduces soil fertility for long
periods due to removal of soil containing essential nutrients. In many cases, extreme drought
conditions precede dust storms (Sivakumar, 2005; Sissakian et al., 2013), as dry soil
disaggregates faster and thus dislodges more easily enhancing erosion. Wind erosion is
thereby closely related to land use practices.

Physiological water stress can be enhanced by increased evapotranspiration, due to high wind
speed. The longer such wind conditions last, the more severe the risk as exemplified by a
recent drought event in India (Masroor et al. 2020). Thus, wind can exacerbate drought
conditions and lead to crop failure. While wind speed is not expected to increase as a global
average (McVicar et al., 2012), evapotranspiration likely will in many regions due to the
increased evaporative demand caused by higher air temperatures (Tomas-Burguera et al.,
2020) and a reduced number of days with rainfall. The fact that some plants react to hot and
windy weather conditions by closing their stomata, may balance some of the enhanced
evapotranspiration deficit. However, this is at the expense of plant growth.

### 4.5.3 Protection measures against wind

Because of the direct wind damage in agriculture, it is necessary or even indispensable to
take countermeasures to minimize the risks. Such measures can be a better choice of location
according to topographic features or using windbreaks. Windbreaks usually consist of natural
barriers such as tree rows. The most important aspect of a windbreak is its height (Brandle et
al. 2004). Indeed, windbreak effects on adjacent crops result in a yield reduction due to water
and light competition up to a distance of one to two windbreak heights, which is followed by a
yield increase up to a distance 8–12 heights (Weninger et al. 2021). To moderate effects of
wind flow around the windbreak, it should be at least ten times as broad as it is high (Brandle
et al. 2004).

### 4.6. Wind-based energy production
Wind indices are of interest for estimating the wind potential and wind energy. Extreme wind
events on different spatial and temporal scales, e. g. storms, gustiness or low-level jets, affect
the energy production, the structural integrity and operational safety of wind turbines.
Microscale variability in the wind field occurs temporally (e.g., gustiness) and spatially (e.g.,
vertical wind shear). These variations of the wind field depend on the time of day and thus on
the stability of the atmospheric stratification. There is also a dependence on the characteristics
of the wind turbine site (land use, terrain). Microscale variations of the wind field influence both
the wind potential and the operational reliability of a wind turbine.

Wind indices are typically defined as the ratio of the current values of a variable to the long-
term mean. The variable is either related to the wind speed or to the wind energy production.
Extreme wind events are directly related to wind speed-based indices. To identify the energy
potential at a site, the Power Density Wind Index can be used (Katinas et al., 2018; Celik,
2003). It is based on parameters of wind speed frequency distribution. The Power Density
Index results in significantly higher variations than the real energy production of the wind
turbine at the location and should be applied carefully. In practice, both the current values of
the wind speed are needed (control of the turbine) and the evaluation of the annual energy
yield compared to the long-term average using wind indices (planning of turbines, financing)

When addressing wind climate at a location, including the occurrence of strong wind events, which includes both productive and destructive events, much attention was given to the connection between the wind climate and the wind energy potential (Carta and Mentado, 2007). In comparison to the wind speed-based indices, the production-based indices use the energy yield of turbines as input data. The Wind Energy Production Index can be based on a Wind Speed Index (Ritter et al., 2015) calculated from wind speed data by an additional application of a power curve (Hahn and Rohrig, 2003; Ding et al., 2005). Another possibility is the use of energy yield data of a wind turbine directly. The BDB index (BDB, 2021) describes the ratio of monthly reported energy yields from wind turbines in a region to the long-term mean yields of these wind turbines. High wind speed or wind shear due to storms or low-level jets need to be taken into account when calculating wind speed indices. However, the energy production-based indices contain the effects of such events only when the wind turbine is working, i.e. until reaching the turbine cut-out wind speed. Due to their design, most systems switch off at a wind speed above 25 m/s (Christakos et al., 2016), but there are also slightly higher and lower shutdown wind speed values for different system types (Chauhan and Saini et al., 2014). An analysis showed that storms had a positive effect on the wind energy production for Southwestern Europe and the Iberian Peninsula (Gonçalves et al. 2020, 2021). As such, the highest values of wind energy production result for stormy weather conditions (Petrović and Bottasso, 2014). Climate change impacts on wind energy have been investigated for a few years (Pryor and Barthelmie, 2010; Moemken et al., 2018). The studies are mostly in agreement on a minimal effect of climate change on the wind energy production (Jung and Schindler, 2020).

Topographic effects are another example of small-scale effects on the wind field, leading to a local wind speed-up, separation, and reattachment. These processes can be studied by numerical models (Uchida and Ohya, 2003, 2008, 2011; Uchida and Li, 2018; Uchida, and Sugitani, 2020). Uchida and Kawashima (2019) defined two indices to evaluate the terrain-induced turbulence and the fatigue damage based on the measurement data and the design value. These studies indicated the need for further development of standards. A commonly used turbulence index is the effective turbulence for site-specific fatigue assessment of wind turbines (Slot et al., 2019). Additionally, the usage of the effective turbulence index significantly reduces the number of aero-elastic simulations needed for checking the loads on major components of the wind turbine.

**4.7. Compound indices**

Strong winds often co-occur with other phenomena and their co-occurrence affects the damage levels observed. This is an integral part of the compound event concept in which multiple phenomena or hazards form a complex causal chain of events that can lead to a more extreme impact than each phenomenon by itself (Zscheischler et al. 2018). A compound event is often associated with one driver (e.g. an extreme cyclone) which may cause multiple hazards (e.g. strong wind and heavy precipitation), but it can have more complex characteristics (Zscheischler et al. 2020). For example, strong wind can also serve as a modulator for hazards like drought and wildfire. A full typology of compound events can be found in Zscheischler et al. (2020).

1012

### 4.7.1 Precipitation.

Strong wind speed often co-occurs with heavy precipitation (Martius et al. 2016), causing multivariate compound events. Additionally, it is argued that wind and precipitation enhance the impact by extratropical cyclones, since cyclones with extreme precipitation often have a longer lifetime than cyclones with only extreme wind speed (Messmer & Simmonds, 2021). Furthermore, the impact of such multivariate compound events is much higher than a hazard containing only wind or precipitation (Martius et al. 2016). In coastal areas, even when wind is not considered as a hazard itself, wind together with heavy precipitation can cause storm surges and coastal flooding (Wahl et al. 2015; Couasnon et al., 2020). Furthermore, precipitation is important when saturating the soil prior to the occurrence of a windstorm. Soil water content is an index that governs the stability of the root sector of trees during storm events (Everham and Brokaw, 1996; Défossez et al. 2021).

### 4.7.2 Air Temperature.

Wind and low air temperatures are both drivers, causing wind chill as human health and agricultural hazards among other risks. Each driver, when acting by itself, would have caused less of an impact than the compound effect (Danielsson, 1996). Wind chill is a threat mainly in cold climates, where enhanced wind speed increases the heat transfer from an object. Such heat loss can cause injuries and mortality both in animals and plants. Windchill can be calculated as the wind chill temperature, also called wind chill factor (Quayle and Steadman, 1998; Bluestein and Zecher, 1999), which is usually taken as the air temperature at which there would be an equivalent rate of heat loss. Also, low air temperatures can lead to the freezing of soil and enhances the stability of trees against windthrow during windstorms (Pasztor et al., 2015). In contrast, trees in frozen soil are more likely to undergo stem breakage than uprooting (Everham and Brokaw, 1996; Peltola, 2006).

### 4.7.3 Drought.
The impact of wind on drought is comparatively small compared with other drivers like temperature and (lack of) precipitation, but it has an effect in terms of the evapotranspiration. Wind is thus included in some drought indices through evapotranspiration in the Penman or Penman-Monteith equation, such as in the Baumgartner index (Baumgartner et al., 1967). These indices are therefore short-term indices that operate on a scale of days and typically do not take into account the long-term impacts of drought on the risk of wind damage to forests. Drought can be considered as a pre-condition, that potentially amplifies the impact of winds. Csilléry et al. (2017) showed that long-term drought can increase the risk of wind damage on sites where drought can lead to a weakening of trees but can also decrease the risk of damage on normally extremely wet sites.

### 4.7.4 Fire.
Indices used for assessing fire risk include often wind and topography to determine the rate of spread and damage caused by a wildfire. Wind and slope are viewed as the major factors influencing fire development (Byram 1959a, 1959b; Sharples 2008). The most used indices for fire risk are based on the Canadian Forest Fire Weather Index (FWI) system (Van Wagner, 1987) that uses information on fuel loading and meteorological conditions (rainfall, temperature, humidity, and wind speed) to predict the probability of a fire starting and then the probable spread of the fire. Humidity, wind speed and air temperature are used to calculate the day-to-day drying of the fuel load. The Initial Spread Index is then used to adjust the FWI as an exponential function of wind speed (doubles the FWI for every increase of wind speed

by 19 km/h or 5.3 m/s). The spread of the fire will also be affected by the topography and, in
particular, how the topography modifies the wind speed and direction.

Wind can alter the angle of the fire toward unburnt fuel, extending the preheating range and
increasing the rate of spread. Slope has a similar effect by affecting the distance between the
flames and the fuel. Thus, typically the greatest rate of spread is found when an upslope is
combined with upward winds and vice versa (Sharples 2008). Since topography influences
wind traits, it can create a channeling effect enhancing fire intensity, but with the strength of
the effect depending on the overlap between wind direction and landscape orientations
(Barros 2012; Mansuy 2014). Kushal (1997) found in a review that a higher relative elevation,
proximity to ridges and increased exposure to wind, all led to greater fire damage in forests.
Additionally, aspects that are associated with greater exposure to dry winds increased fire
damage in forests, and damage was lower in aspects with cold and moist winds. There is an
index combining slope, aspect, and wind speed (wind-topo), but it had a rather low importance
for the final model chosen for statistical interpretation (Masoudvaziri 2020).

**4.8.    Wind speed warning-levels used at national meteorological services and sector-**
**related critical thresholds**
Advanced storm-warnings are crucial for the protection of property and lives. Meteorological
services operate a structured warning system for windstorms and recommend appropriate
protective measures and rules of conduct depending on the warning level (e.g. Germany:
(DWD, 2021), Ireland: (MetEireann, 2023) or Sweden: (SMHI, 2023)). The warnings will be
published, when the event reaches a certain probability level to occur, can be well spatially
located and especially when the warning criterion is met, such as wind speed or precipitation
exceeding a certain threshold value. These threshold values are set individually by all
meteorological services. In some cases, the weather services already indicate possible
consequences due to the wind speed, by warning of damage to infrastructure, forests, or
energy systems at differing warning levels. There are even variants of weather forecasting
systems that follow a more risk-based approach, i.e. the probabilities and consequences of
extreme events are integrated into the forecasting system in order to achieve an improved
warning management (Neal et al. 2014 or Kaltenberger et al. 2020). A Europe-wide overview
of warnings and, in part, possible impacts is provided by Meteoalarm (www.meteoalarm.org),
developed by EUMETNET (European Meteorological Network) provides relevant information
on extreme weather events from 37 national meteorological services.

We collected many critical thresholds from the literature for the five sectors which are the focus
of this manuscript. The vulnerability of each sector to wind speed is illustrated in Table S1.
Figure 3 provides a synthesis and comparison of thresholds from the five sectors. The
agriculture sector seems to be the most sensitive to wind, as negative effects are already
noticeable at mean wind speed well below the first official warning level of the DWD (WL1 =
14 m/s; Rouse and Hodges, 2004).

At WL2 (18-29 m/s), initial restrictions must already be expected in all 5 sectors, but these are
initially localized. In the forest, individual trees and areas may be affected (Gardiner et al.,
2010, 2013, 2016), buildings may show slight roof damage but no structural damage yet
(Feuerstein et al., 2011), in road traffic there may be some accidents and delays in train and
air traffic (Vajda et al., 2014). Concerning wind energy, no damage is expected yet, but
depending on the type of turbine, precautionary shutdowns of turbines may occur. For WL3
(29-39 m/s), the literature describes significant impacts in the forest, building, and
transportation sectors. Damage will be significant, and impacts are already affecting regional
areas. The influence of storms on transportation can quickly impact society at regional to
national levels.
Severe damage is described at the national level from WL4 onwards, including damage to
wind turbines (Quaschnig et al., 2016), massive building damage (Feuerstein et al., 2011), or
even the shutdown of entire transport sectors (air and railway) (Vajda et al., 2014). While
forest, urban areas and transport are affected by wind speed at the same order of magnitude
(i.e. consequences for society are mostly locally at WL2, regionally at WL3 and nationally at
WL4), the energy production from wind is impacted at a much higher wind speed (when only
damage is considered) while agricultural productivity at much lower wind speed.

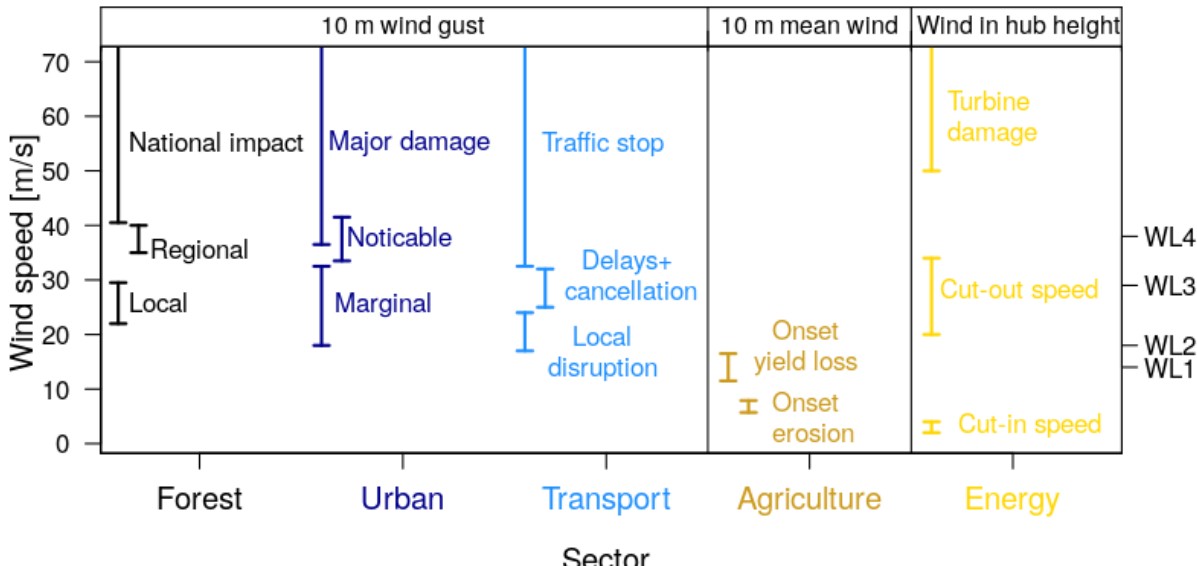


Figure 3: Critical threshold ranges of wind speed (mean wind speed (averaging interval 1 hour)
and wind gusts) for five affected sectors. Warning levels (WL1-4 at speeds of 14, 18, 29 and
39 m/s, respectively) of the DWD for wind gusts (Primo, 2016) are marked on the right axis.
For each sector different ranges of critical mean wind speed from the literature are plotted, to
show the mean wind speed (or gust), where impacts are expected. Thresholds in the first three
sectors (forest, urban areas and transport) refer to wind gusts at 10 m height, for agriculture
we present the mean wind speed at 10 m height and the shown thresholds for the energy
sector refers to the mean wind speed measured at the height of the wind turbines (wind in hub
height). The upper bar (e.g. "National impact") is left open as damage will occur also at higher
wind speed. The threshold ranges mean: (1) Forest local: Limited area of damage. Forest
Regional: Damage level is meaningful for the affected forest and short-term forest planning
and timber price. Forest National: Damage can occur across several countries. (2) Urban
Marginal: Light objects, tiles can be lifted or come loose. Urban Noticeable: Heavier objects
are lifted and first damage to individual building components are possible. Urban Major: Large
vehicles overturn, roofs are severely damaged. (3) Transport Local disruption: Blocked roads
through windthrow or sliding containers at ships. Transport Delays: Cancellation trough
electricity cuts and increasing number of wind-related accidents. Transport Traffic stop:
Damage of overhead cables and longer power failures as well as airport and harbor closures.
(4) Agriculture erosion: Soil loss to wind erosion. Agriculture yield loss: Damage to leaves and
yield loss due to lodged fields. (5) Energy cut-in speed: start of energy production.  Energy
Cut-out speed: Automatic shutdown of wind turbines.

## 5.      Outlook & open research questions

In this review we covered a wide range of topics dealing with wind damage to terrestrial ecosystems with an emphasis on studies dealing with Central Europe. To conclude, we address issues of importance in the near future and topics that require further research. The most intriguing question in this field is how wind-related damage levels may change in future decades, given the strong dominance of decadal variability (Feser et al., 2015). Therefore, attention was given to identifying drivers of future changes in windstorms and cyclone characteristics which are particularly important for the predictability of present-day and long-term trends in socio-economic damage (Koks and Haer 2020; Hoeppe 2016; Franzke 2021). The key current drivers that contribute to future changes in storms are well known; many studies assume that the atmospheric moisture content will increase due to global warming (IPCC 2021). Idealized studies suggest that this increase in moisture will lead to a stronger circulation, more intense storms (including stronger winds and more rainfall) and, thus, to an expansion of the windstorm footprint (Catto et al. 2019). Additionally, studies show that the lower-tropospheric meridional air temperature gradient will decrease due to Arctic amplification, whereas the upper-tropospheric meridional air temperature gradient will increase due to the warming of the tropical upper troposphere and the cooling of the polar lower stratosphere (Lee et al. 2019). However, it is still uncertain how these contrasting forcing mechanisms will contribute to the future changes in storms quantitatively (Catto et al., 2019; their Fig. 2). The recently extended ERA5 reanalysis product could enable further studies to deal with wind-related damage in the past, present and future, reducing uncertainties. Indeed, increasing the resolution of climate models may improve their capacity to quantify statistical storm properties. CMIP6 models already indicate a general improvement in future storm tracking (Priestley et al. 2020b; Harvey et al. 2020). As a result, more accurate projections of wind and storm damage based on future emission scenarios and climate change may be attainable in the future. According to a recent study, winter storm-related wind gusts could increase towards the 2nd half of the 21st century in Germany (Jung und Schindler 2021). This demonstrates the need for more studies in damage analysis.

Methodologically, the usage of the indices described here in damage analysis has many advantages, but their creation can be time consuming, and their usage may lead to statistical pitfalls. For example, it is important to choose indices while trying to avoid an overlap in variability explained by different topographic indices (Mitchel 2001). In this sense, we are lacking a clear methodology that can select the most suitable indices in advance, especially when many indices are easily available. There are three main approaches: 1) a hypothesis-based approach, where typically only few variables are used in the analyses because these variables can be well explained and justified due to past research, incorporation of expert knowledge in the development of indices using co-design and familiarity with the study site (Gebhardt et al. 2019, Merz et al. 2020), 2) a computational approach, where a feature-selection algorithm (e.g. genetic algorithm) is first used to trim down the number of independent variables before performing an analysis, and 3) an exploratory approach with little limitation on the number of independent variables used, where one can examine, for example, if a certain group of indices is more useful than another (e.g. gust-related indices vs. topographic indices) in achieving accurate models according to a given evaluation metric (e.g. coefficient of determination or area under the curve).   The choice of method is dependent on the specific research goals, but also on the skillset and computational resources available, for instance, an exploratory analysis including many variables on a large area may demand access to high-performance computing. Furthermore, when modelling on a large spatial scale,

it is important to choose analysis tools that test and quantify the homogeneity of the relation
between indices and damage variables across the different sub-regions in the study site. Thus,
taking into account that key parameters may change within the study area, such as the
topography or the vegetation structure, altering the relations between an independent variable
and storm damage. Finally, when analyzing socio-economic impacts, the availability of data is
often a limiting factor, and these limitations shape the selection and analysis approach.
We identify that the area most in need of new indices for wind-related damage analysis are
compound events. Damage from extreme climatic events most commonly occurs through
interactions between different hazards (Zscheischler et al., 2020). The main challenge is to
handle the different time scales of each factor, for example, a storm may last from hours to
days, but drought can last years. Therefore, we require indices that incorporate a multitude of
factors that are very site specific, as both the topography and the land cover can strongly
modify these interactions. Another important challenge is the inclusion of non-climate drivers
related to exposure and vulnerability in the compound indices. Concerning the five sectors
dealt with here, we present sector-specific outlooks:

**5.1 Forest.** In a forest setting, there are very few measurements of tree damage due to storms
(Kamimura et al., 2022) and very few studies of the dynamic nature of damage at the time
scale of a storm. Such studies are required to understand damage initiation and propagation
during storms (Dupont et al., 2015). In addition, predicting airflow over complex terrain is still
difficult when there are steep slopes and multiple changes in vegetation height (Finnigan et
al., 2020). Similarly, there is a need for improvement of land surface information, and in
particular, the acquirement of highly resolved 3D distributions of vegetation elements at the
landscape scale to enable the creation of fine scale maps for risk assessment. To this end, it
is often difficult to assess damage or risk in the most relevant spatial scale. The recent
developments in remote sensing techniques (terrestrial and airborne laser scanning) promise
effective assessments of surfaces structures (Favorskaya and Jain, 2017), may prove useful
for many of these issues. However, it may take much time to achieve a sufficient level of data
collection, for example, terrestrial laser scanning is accurate but confined to small areas, and
an effective assessment for larger areas using airborne laser scanning and satellite data are
not at a sufficient resolution and need further development. Another consequence is that we
still lack in monitoring and modelling the small-scale variability in the interactions of the wind
field with the surface. The main research questions for the future are: How does the structure
of a forest canopy influence the turbulence within and above the canopy? And, as they grow,
stems, roots and canopies acclimate to the wind forces, so, what is the optimal cultivation and
canopy structure to reduce damage (Dèfossez et al., 2022)?
**5.2 Urban.** In urban settings, storm, and loss indices as well as damage functions do not
usually consider differences in the exposure and vulnerability of different building types or
types of urban areas. To further assess wind damage risks on a smaller spatial scale
investigations of individual building damage or damage to specific types of neighborhoods are
needed, together with modelling of urban areas. However, damage data at a fine spatial scale
is difficult to obtain, and it is a priority to improve the documentation of urban damage to
support the development of new indices. The availability of data, such as the spatial extent of
wind damage to individual buildings or green spaces, is key in developing mitigation strategies.
For example, wind channeling as a function of wind speed and direction needs to be reliably
simulated during the development phase of new building projects.
**5.3 Transport.** Studies addressing wind effects on transport usually focus on direct effects in
a particular part of the transport system. Results from such studies can strongly depend on
the region, data and methodologies used for the study. Studies with a more unified approach
addressing wind effects on transport on a broader scale could lead to more comparable
results. Furthermore, little research is available that takes into account cascading effects that
propagate through different parts of the transport system. In general, it remains unclear how
climate change and resulting changes in the wind extremes will affect the transport system.
Studies addressing this question should not only consider future changes in wind extremes,
but also potential changes of the transport system as part of climate change mitigation
measures. Such measures could make the transport system more vulnerable to extreme
winds. For example, a shift from road to rail transport to reduce $CO_2$ emissions could lead to
a higher vulnerability to wind-related tree fall, because single storm events can lead to a
collapse of rail transport over whole countries for periods of several days.
**5.4 Agriculture.** The future of the agriculture sector is closely linked to the global challenge
of feeding a still growing population, which is expected to reach 9.7 billion people by 2050 (UN
World Population Prospects 2022). In response to this challenge, the awareness for
sustainable and efficient agricultural practices has been gradually increasing. Wind damage
in agriculture landscapes is thereby a growing concern due to the potential change in
frequency and intensity of wind events as the climate continues to warm (Seneviratne et al.
2021). In order to optimize crop yields and reduce waste, the relationship between wind
damage and crop yields needs to be investigated in more detail to quantify this impact. For
example, understanding and better short-term prediction of wind events are key to improved
crop management. Furthermore, there is a lack of simple indices incorporating soil properties
and their tendency to lead to soil erosion and nutrient loss, or wind erosion, as such events
can be a major challenge for farmers. There is much space to develop new practices to
mitigate wind damage in agriculture by using vegetation and cover crops reduce wind damage.
Since the positioning of vegetation (e.g. trees as windbreaks) alter the small-scale interactions
of the wind field with the soil and crops, a more accurate positioning of vegetation would be
supported by the creation or adaptation of existing compound indices or modeling platforms.
Such manipulations of the surface cover can be highly flexible in the spatial scale of the wind-
field modification, thus providing a good counter measure to different types of vulnerability in
agricultural sector.
**5.5 Renewable Energy.** It is important to follow the influence of climate change projections
on wind energy production. With the increase in the reliance on renewable energy, it will be
important to reduce uncertainties in wind potential and the risk for technical and safety issues
in the operation of the wind turbines. Furthermore, while we know where turbines are located
and their characteristics, it would be important that the data on the turbines wind field and the
energy generated were accessible for scientific projects and to the private sector. Currently
much of the data is not made publicly available. For instance, we especially lack wind data at
hub height for the evaluation of numerical models. Other key challenges, that are similar to
other sectors, are the acquirement of high spatial resolution measurement, and past and future
modeling of the wind field over heterogeneous surfaces and complex terrain.

In conclusion, predicting and assessing the damage caused by wind and storms is a complex matter but there are effective and simple methodologies to support assessment and decision making. In the light of future uncertainties, it is vital to continue developing tools to prepare for the next calamities that are bound to occur.

**6. Acknowledgements:**
This publication is the outcome of a working group of the project ClimXtreme, funded by the German Bundesministerium fuer Bildung und Forschung (AK, DG, DH, DN, HL, JG). AZ was funded by Project QuWind100 (German Federal Ministry for Economic Affairs and Energy, BMWi, based on a decision by the German Bundestag, funding code 0325940A). CF was supported by the Institute for Basic Science (IBS), Republic of Korea, under IBS-R028-D1 and by National Research Fund of Korea (NRF-2022M3K3A1097082). JGP thanks the AXA Research Fund for their support. NB's work was carried out within the framework of the Hans-Ertel-Centre for Weather Research. This research network of universities, research institutes and the Deutscher Wetterdienst is funded by the Bundesministerium für Verkehr und Digitale Infrastruktur (grant no. 4818DWDP3A). JGP is a member of the editorial board of the journal Natural Hazards and Earth System Sciences.

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

2241   o