# Peer review of "Review Article: A European Perspective on Wind and storm damage: From the meteorological background to index-based approaches to assess Impacts"

_Natural Hazards and Earth System Sciences, 2022_

## Author Comment (AC1)

Point-to-Point Response to Reviewer Comments

Reviewer #1

Thank you very much for reviewing our manuscript and your helpful and very detailed comments; they helped us to considerably improve the manuscript.

**1) Reviewer comment:** According to the title, this manuscript addresses the topic of wind and storm damage from the genesis of meteorological conditions that lead to storms to the damage that storms can cause. In my view, the title of the manuscript is too general, as the authors focus on extra-tropical storms and their impacts over the North Atlantic-European region. One can clearly see that the review was written from a European perspective. Therefore, I suggest a title change that more closely reflects the focus of the manuscript and the spatial relevance of the summarized studies. An imbalance can also be observed concerning the five sectors studied. The text portions in which wind effects on forests and trees are described predominate. This is not unfavorable in principle but should be clarified in the introduction.

**Our response:**

We changed the title to "A European Perspective on Wind and storm damage: From the meteorological background to index-based approaches to assess Impacts" to clarify our predominantly European perspective. We also added to the introduction that we put particular emphasis on wind effects on forests and trees "We focused on five key sectors: forests, urban areas, transport, agriculture, and wind-based energy production. For each sector we described indices and thresholds relating to physical properties such as topography and land cover but also to economic aspects (e.g. disruptions in transportation or energy production)" and "Here, we aim to bring these different disciplines together to provide an interdisciplinary synthesis of the topic. To bridge the gap between the different communities, within the ClimXtreme consortium, we created a work group and invited specialists from outside the consortium to broaden our research expertise. During regular joint meetings we identified the following sectors: forests, urban areas, transport, agriculture, and energy as the most relevant terrestrial environments that could be impacted by wind and storm damage. We focused on literature resources stemming mainly from Europe, but in cases of relevance and to further expand the scope of the review we also incorporated examples from other regions".

See lines 45-49, 109-117.

**2) Reviewer comment:** I suggest providing definitions (e.g., based on numerical values, duration, ...) of "wind", "windstorm", "storm", "wind-related risk", "gust", "mean wind speed", ... early in the Introduction. In the context of "wind-related risk assessment" this is important. What is "wind-related risk"? Is this wind and storm damage risk? Not all wind speed values pose a risk and cause calamities.

**Our response:**

We define wind as "Wind is per definition a sustained air movement in the atmosphere". See lines 54-56.

We define windstorm as "Windstorms produce winds which strong enough to cause damage; they typically have windspeeds in excess of 15m/s." See lines 216-217.

We define storm as "storms, which have very strong winds". See lines 57-58.

Since we use "wind-related risk" just once, we instead define "risk" as "By risk we understand here the likelihood that wind causes some damage and their consequences." See lines 83-85.

Wind gusts: "Wind gusts are sudden increases in windspeed, which lasts typically less than 20 seconds." See lines 68-70.

"Mean wind speed" is defined as "the wind speed averaged over some time period". See line 685.

**3) Reviewer comment:** I suggest deleting "e.g." when citing other studies (e.g. P2L57 "e.g. Bittelli et al., 2008) throughout the text.

**Our response:**

Done.

**4) Reviewer comment:** I suggest homogenizing the citations style in the text. Please pay attention to the use of the comma and semicolon.

**Our response:**

Done

**5) Reviewer comment:** Please homogenize and correct the order and formatting of all citations in the text. I found different formats and orders (by authors vs by year).

**Our response:**

Done.

**6) Reviewer comment:** Please check the formatting and grammar of all headings.

**Our response:**

We have improved the headings throughout the manuscript.

**7) Reviewer comment:** Please insert a blank character between numbers and units wherever it is missing.

**Our response:**

Done.

**8) Reviewer comment:** Please add more line breaks where new trains of thought begin.

**Our response:**

We went through the whole manuscript and made changes.

**9) Reviewer comment:** I suggest a complete review of the formatting of abbreviations and symbols. Particular attention should be paid to the subscript of letters.

**Our response:**

Done.

**10) Reviewer comment:** It seems as if sections were written by different authors. There are discernable differences in the diction, and other inconsistencies:

**Our response:**

Yes, all co-authors contributed to the manuscript, with all having their own writing style. We went over the entire manuscript and homogenized the style as much as possible.

**11) Reviewer comment:** Please make sure that there are no repetitions of definitions or basic facts in different sections. For example, "wind load" sure be defined once at the very beginning, not in sections 4.3.

**Our response:**

"Wind load" is now only once defined, at its first use.

**12) Reviewer comment:** The sections have a differing structure of subsections. I suggest using the same or a similar structure and contents in the sections. This considerably increases the readability of the long manuscript.

**Our response:**

We have restructured our subsections so that have now a systematic numbering structure.

We are not sure what the reviewer means by "similar content" in the sections. Since we cover a diverse set of wind related topics, and those topics determine the contents of our sections and subsections. We also did our best to further homogenize the text.

**13) Reviewer comment:** I suggest moving all knowledge, facts, concepts, and descriptions (e.g., structure of the lower parts of the boundary layer, wind pressure, vertical wind profile, coherent structures, roughness, porosity, channeling, codes, terrain roughness, orography index vs topography index, ...) that are portable to all "environments" are provided in separate sections before presenting the specifics of the environments.

**Our response:**

We are not sure we understand your comment. We went over the whole manuscript to streamline its structure and flow. We hope that this will also have addressed this comment.

**14) Reviewer comment:** Please check the entire text for the incorrect use of uncountable nouns such as damage and wind speed.

**Our response:**

Done.

**15) Reviewer comment:** L43: What is the difference between wind damage and storm damage? What is the metric that is used to distinguish the two kinds of damage?

**Our response:**

We now define both in the introduction: "With storm damage we refer to damages, mainly to properties and forests, caused by severe wind storms, while wind damage is more general and includes all adverse effects of wind, including storm damage." See lines 80-85.

**16) Reviewer comment:** L50: I suggest using the correct technical term "air temperature" instead of "temperature".

**Our response:**

Done.

**17) Reviewer comment:** L66: What is the difference between "strong winds" and "strong wind gusts"? I suggest providing clear definitions of both quantities.

**Our response:**

We now define both: "Wind gusts are sudden increases in windspeed, which lasts typically less than 20 seconds, while strong winds refer to sustained wind speeds over longer time periods". See lines 68-70.

**18) Reviewer comment:** L67-L75: At this point it would make sense to introduce the definition of risk. The storm damage risk is the product of hazard, exposition, and vulnerability.

**Our response:**

We introduce risk now at this location: "We define risk as the likelihood that wind causes some damage and their consequences and risk can be quantified as the product of hazard probability, exposure and vulnerability (e.g. Kelman 2003)." See lines 83-85.

**19) Reviewer comment:** L75: I suggest better structuring and formulating the list "wind, storm dynamics, and the ability ..." What is the meaning of "understanding wind"?

**Our response:**

Changed sentence to **"**The understanding of wind, storm dynamics, and the ability to predict the damage they cause, requires an interdisciplinary approach". See lines 107-108.

**20) Reviewer comment:** L81: Do you solely mean "wind damage"? Or also storm damage? I suggest replacing "wind-damage" with "wind damage".

**Our response:**

Agreed. "wind-damage" changed to "wind damage" throughout. In this paper we focus on wind damage and not other types of damage

**21) Reviewer comment:** L82-L86: I suggest deleting all repeated information that has already been provided in previous lines.

**Our response:**

We deleted repeated information.

**22) Reviewer comment:** L92: Please replace "in the Earths'" with "of the Earth's".

**Our response:**

Done.

**23) Reviewer comment:** L103: Please delete the superfluous "below".

**Our response:**

Done.

**24) Reviewer comment:** L110: I suggest lower casing "Northern", "Southern" and "Hemisphere".

**Our response:**

Done.

**25) Reviewer comment:** L117-L123: These lines are located between explanations about the jet stream. Therefore, I suggest moving these lines further down and to merge the information about the jet stream. It would also make sense with respect to the space and time scales that boundary layer processes dominate to move the lines downward.

**Our response:**

We moved this paragraph down. See lines 179-185.

**26) Reviewer comment:** L136-L144: It is not clear why these explanations are provided. I doubt that the entire readership knows how "vorticity" is defined. Where is the connection to air motion in the context of this paper?

**Our response:**

We have removed the sentences mentioning vorticity.

**27) Reviewer comment:** L147: I suggest replacing "its" with "their".

**Our response:**

Done.

**28) Reviewer comment:** L195: Please adjust the formatting of the heading to the formatting of the other headings.

**Our response:**

We have homogenized all headings.

**29) Reviewer comment:** L200-L201: I suggest deleting the lumped references. They are repeated in the following lines.

**Our response:**

We have deleted the references which are repeated in the next paragraph.

**30) Reviewer comment:** L206: "CMIP5" is undefined. I suggest providing a definition.

**Our response:**

We now define CMIP at first use. See lines 253-254.

**31) Reviewer comment:** L211: Please correct the heading's formatting. Why are "Circulation Characteristics" capitalized?

**Our response:** We have changed "Circulation Characteristics" to "circulation characteristics"

**32) Reviewer comment:** L222: Please provide a definition of abbreviation "ERA5".

**Our response:**

We now define ERA5 at first use. See lines 270-271.

**33) Reviewer comment:** L223: Please homogenize the information on the geographical coordinates.

**Our response:**

Done

**34) Reviewer comment:** L230: I suggest lower casing "Sea Level Anomalies".

**Our response:**

Done

**35) Reviewer comment:** L232: Please insert a blank character between "250" and "hPa".

**Our response:**

Done

**36) Reviewer comment:** L234: Is it necessary to display the sequence of weather regimes (CL1-CL5)? I suggest deleting Fig. 1c. It does not provide information needed for this review.

**Our response:**

We deleted Fig. 1c and adapted the text of the figure caption of Fig. 1, and the related paragraph.

**37) Reviewer comment:** L235: Is "red arrow" correct? I do not find a red arrow.

**Our response:**

Since we deleted fig. 1c, no arrow is displayed in fig. 1 any longer now.

**38) Reviewer comment:** L235: What is "eXtreme WindStorms"? Is this an expression needed for this review?

**Our response:**

This refers to the XWS data base which we now explicitly state and reference. See line 285.

**39) Reviewer comment:** L236: Please replace "storm track for storm" with "track for storm" or "storm track of Klaus".

**Our response:**

We replaced "storm track for storm" with "track for storm". See line 285.

**40) Reviewer comment:** L235: Please add the missing dot after "al".

**Our response:**

Done

**41) Reviewer comment:** L229-L256: Please homogenize the use of quotation marks when referring to winter storm Claus.

**Our response:**

We removed the quotation marks.

**42) Reviewer comment:** L244: Please replace "5" with "five".

**Our response:**

Done.

**43) Reviewer comment:** L252: The citations "Liberato et al., 2011" can be deleted. It is used again in L256.

**Our response:**

Done.

**44) Reviewer comment:** L254, L262: Is "jet-stream pattern" the same as "jet regime", as used on P6L232? If so, I suggest homogenizing the technical terms to minimize the load of specialized speech.

**Our response:**

The shown jet-stream patterns have been determined as those patterns which are associated with the circulation regimes determined form the sea-level pressure (SLP)  data by compositing the zonal wind anomalies at 250 hPa over the days assigned to each SLP regime. Therefor we agree that "jet regime" is misleading here and homogenize the terms:

"The patterns for the associated jet regimes" --> "The jet-stream patterns associated with the individual weather regimes". See lines 282-284.

"The 5 jet regimes" --> "The five jet-stream patterns". See line 293.

**45) Reviewer comment:** L258-L262: When looking at Fig. 1c, I find many CL4 regimes where no severe storm has occurred. In my opinion, this considerably relativizes the statements made here regarding specific weather regimes. I therefore suggest deleting or at least reformulating these lines.

**Our response:**

We removed Fig. 1c.

**46) Reviewer comment:** L264: Please check the grammar. Is this the right case? Please lower case "Seasonal Variability".

**Our response:**

We changed the heading to "Temporal characteristics of storms and seasonal variability". See line 313.

**47) Reviewer comment:** L270-L271: What is a "wind interval"? Do you mean "wind speed interval"? Please clarify.

**Our response:**

Indeed, wind speed interval is meant and corrected. See line 320.

**48) Reviewer comment:** L272-L273: The reference of the relative pronoun is ambiguous. Please considering rephrasing this sentence.

**Our response:**

We rephrased it. See lines 324-325.

**49) Reviewer comment:** L274: There is no definition of "WL3" and "WL4". Please provide more details.

**Our response:**

We moved the information from chapter 4.8 here, to define the used WL. See lines 321-324.

**50) Reviewer comment:** L282-L283: Please provide correctly formatted tildes.

**Our response:**

Done

**51) Reviewer comment:** L286: Please homogenize the spelling of "intraseasonal". What is your definition of "intraseasonal"? For me, this is the variation of wind speed within one season, e.g., winter. Is this the same definition for you? Or do you mean "intra-annual", which is the variation of wind speed between seasons over the year. Please clarify.

**Our response:**

The reviewer is right. There was a mix-up of words. We replaced it by intra-annual. See line 339.

**52) Reviewer comment:** L312: Why is "speeds" provided in km/h? Previous "speed" values were provided in m/s? Please homogenize. Which "speeds" do you mean? Wind speed or gust speed or anything else? Please clarify.

**Our response:**

We state now that we mean wind gusts and changed to m/s.

**53) Reviewer comment:** L314: Please check the double use of "from".

**Our response:**

We rephrased this. See line 370.

**54) Reviewer comment:** L316: What do you mean by "commonly". Can this be quantified?

**Our response:**

We have rewritten this part and added a reference for this point: "Forest damage from thunderstorms from in areas, which previously were rarely affected, such as eastern parts of Europe (e.g. Nosnikau et

al., 2018; Sulik and Kejna, 2020), but have experienced an increase in convectively available potential energy and near surface moisture which can cause more thunderstorm activity (Taszarek et al. 2021)." See lines 370-375.`

**55) Reviewer comment:** L317: Which "type of convective activity" do you mean? And why is the reference Diffenbaugh et al. (2013) older than the references that refer to "more commonly reported". This is not stringent.

**Our response:**

We added two newer references: Lepore et al. 2021 and Taszarek et al. 2021. See lines 374-375.

**56) Reviewer comment:** L328: What are "wind field" characteristics that are relevant for this review besides "speed". Please be more specific.

**Our response:**

We have rewritten this: "The characteristics of the wind speed and gustiness in a given environment …". See lines 386-388.

**57) Reviewer comment:** L329-L330: Please delete "caused by wind". It is redundant.

**Our response:**

We removed "by wind".

**58) Reviewer comment:** L335. Please homogenize the spelling of "earth" throughout the text.

**Our response:**

Done.

**59) Reviewer comment:** L351: Please homogenized the spelling of "wind speed" vs "wind-speed" in the text.

**Our response:**

Changed throughout.

**60) Reviewer comment:** L367-L377: There are still no generalizable findings on whether the effect of individual tree stability or that of collective stand stability better protects forests from storm damage.

Kamimura et al. (2022, listed in the References) provide important insights in this regard. They should be mentioned here.

**Our response:**

Agreed. We added on line 441 the sentence "Recent experimental measurements of tree damage during a super typhoon (Kamimura et al. (2022) has also shown that collisions between the crowns of individual trees and the crowns of their neighbours is extremely important in reducing tree movement during strong winds and contributing to their overall stability"

Kamimura, K., Nanko, K., Matsumoto, A., Ueno, S., Gardiner, J. and Gardiner, B.: Tree dynamic response and survival in a category-5 tropical cyclone: The case of super typhoon Trami, Sci. Adv., 8(March), 1–11, 2022.

**61) Reviewer comment:** L404-L405: The mention of the five environments is redundant. They have been mentioned before. I suggest deleting them.

**Our response:**

We removed the text as requested.

**62) Reviewer comment:** L409: Please homogenize the spelling of "winter-storm" vs "winter storm" in the text.

**Our response:**

Done.

**63) Reviewer comment:** L472: I suggest replacing "asl" with "above sea level" or defining "asl".

**Our response:**

We exchanged asl with above sea level as requested

**64) Reviewer comment:** L78: I suggest replacing "altitude" with "elevation". While altitude is the height above ground, elevation is the height above sea level.

**Our response:**

Done.

**65) Reviewer comment:** L491: I suggest replacing "damages" with "damage" because it is an uncountable noun in this context.

**Our response:**

Done.

**66) Reviewer comment:** L497, Figure 2: Please replace "usefull" with "useful" in the figure legend.

**Our response:**

Done.

**67) Reviewer comment:** L503: Please provide more information on the critical wind speed. Is this the instantaneous wind speed as sum of mean wind speed and gust speed or is this the mean wind speed alone? If so, what is the averaging interval for CWS?

**Our response:**

The typical averaging interval for CWS is a minute period, e.g,. 3 minutes (Peltola and Kellomäki, 1993) or 10 minutes (Dupont et al., 2015, Peltola et al. 1999). See line 609.

CWS is estimated either at a height of 10 m above the canopy or at the tree top at the stand edge. See lines 611-612.

These CWS are either computed at a height of 10 m above an open lawn surface or at the top of the edge of trees at risk.

We have made these changes.

**68) Reviewer comment:** L510: The maximum bending moment at the stem base is also strongly dependent on the crown architecture and dimensions. Trees with large crowns dissipate large amounts of energy in the crown space before any moment can be measured at the stem base.

**Our response:**

BMmax calculation refers typically to mean bending moment (BMmean) and a gusting factor (see e.g., Gardiner et al. 1997). A tree uproots if its BMmax at the ground level exceeds the resistance of the root–soil plate, and a tree breaks if its maximum bending moment at breast height (1.3 m) exceeds the critical value of the stem's modulus of rupture (Peltola et al. 1999). The gusting factor is parameterised by wind measurement (field or wind tunnel) and the spacing/height ratio of tree stands and depends directly on DBH and indirectly on stem density (Gardiner et al. 1997). The stem density depends also on crown size. The wind measurements were provided at top of the trees, and the bending moment was typically determined in the level of zero displacement (e.g., 0.8 of the tree height, (Gardiner et al. 1997)). Neverless, measurements of the effects (Gardiner et al. 1997) as well as directly solved finite elements models of the crown architecture (Ruy et al., 2022) have shown the influence of crown architecture on

the maximum bending moment. Therefore, different authors recommended the application of adapted parameterisations of BMmax by gusting factor for different stand compositions and tree types (Gardiner et al., 1997).

We have made these changes. See lines 613-630.

**69) Reviewer comment:** L519: Please define the abbreviation "DBH".

**Our response:**

We now define DBH in the text. See line 640.

**70) Reviewer comment:** L525: Please define the abbreviation "NDVI".

**Our response:**

We now define NDVI in the text. See line 647-648.

**71) Reviewer comment:** L529: Please replace "drag coefficient" with "mean drag coefficient". The instantaneous drag coefficient of trees under wind loading is still largely unknown because it varies instantaneously.

**Our response:**

We have changed it in the text. See line 651.

**72) Reviewer comment:** L548: Here, you use "diameter of breast height" instead of "DBH". It would better to consistently used either the long name or the abbreviation.

**Our response:**

We have changed it in the text. See line 672.

**73) Reviewer comment:** L551: Here, you use "Critical wind speed". It would better to consistently used either the long name or the abbreviation.

**Our response:**

We have changed it in the text. See line 675.

**74) Reviewer comment:** L552-L553: Please correct the formatting of the citation.

**Our response:**

Done.

**75) Reviewer comment:** L555-L564: How are these lines connected to wind-forest interactions? I suggest moving these lines into a more general statements section or deleting them.

**Our response:**

We moved the paragraph to "4.1 General storm indices and severity indices". See lines 490-500.

**76) Reviewer comment:** L566: I suggest replacing "Urban" with "Urban areas". "Urban" is unspecific.

**Our response:**

We exchange "urban" with "Urban areas" as requested, here and throughout the manuscript.

**77) Reviewer comment:** P15, section 4.3.: Why are the subsections not numbered? In the previous sections, the subsections were numbered.

**Our response:**

We followed the suggestion of the reviewer and all subsections are now numbered.

**78) Reviewer comment:** There is an inconsistency in the headings of the subsections. The heading of the first subsection in 4.2. was "4.2.1 Topographic indices" (à 4.2.2 Topographic indices). In this section, it is "The urban boundary layer". I suggest, adding a section something like "4.1.2. The forest boundary layer". This would strengthen the structure of the manuscript. Topographic indices are of a totally different quality when it comes to the assessment of "The small-scale interactions ..." (L568), in forest and urban areas.

**Our response:**

We have redone all section headings.

**79) Reviewer comment:** L594-L598: Please replace "damages" with "damage". This is an uncountable noun in this context.

**Our response:**

Done.

**80) Reviewer comment:** L608: What is the difference between "topographic index" and "orography index"? I guess, there is none. I nonetheless suggest homogenizing these technical terms. More semasiologically correct would probably the use of "terrain index".

**Our response:**

We followed the suggestion and changed the text. We prefer to stay with "topography" as it is more frequently used in our literature than "terrain". See line 723-724.

**81) Reviewer comment:** L611: Does "critical wind speed thresholds" correspond to "CWS"? If not, this should be made clear.

**Our response:**

CWS and $v_{crit}$ are in so far similar that one is "the critical wind speed at which buildings suffer damage" and the other is the "threshold wind speed […] to stem breakage or uprooting of a tree". Thus, both describe a wind speed threshold for damage but for different types of damages. They cannot be used interchangeably. CWS is a standing term in forest ecology while it is not used in building damage modeling. Therefore, we would keep the declination as it is. This is now stated in the text. See line 727.

**82) Reviewer comment:** L621-L622: Before, it was stated that gust speed is the most essential factor for storm damage. Here, it is stated that the "maximum daily wind speed" is most influential. Is gust speed equivalent to maximum daily wind speed? If not, what is the difference? What is the period of maximum daily wind speed? Please clarify.

**Our response:**

We changed "wind" to "gust". "daily maximum" refers to the maximum within a day, usually the 24 hours from 00 UTC to 00 UTC of the following day. See line 738.

**83) Reviewer comment:** L645: Please replace ";" with ":".

**Our response:**

Done.

**84) Reviewer comment:** L645-L652: Please match the variables and their definitions listed in Table 2 with variables mentioned elsewhere in the text. Otherwise, provide clearly distinctive definitions. For example: Does $v_{crit}$ correspond to CWS? If so, does CWS have a daily resolution? In the text, you

mentioned "maximum daily wind speed". Does it correspond to $v_{max}$, being the abbreviation for maximum daily gust speed? What is the definition of gust in this context?

**Our response:**

For CWS and $v_{crit}$ see answer under 81). We did not find any other corresponding variables except mean daily wind speeds and maximum daily wind speed. We introduce the reoccurring parameters like maximum daily wind speed, mean daily wind speed, gusts and gust speed in chapter 1.

"For wind indices and wind impact models different wind parameters are in use. These are often derived from modeled data like reanalysis datasets. While these model parameters are strongly related to observed wind parameters, they are not the same and their definitions cannot be used interchangeably. Since observational data is rare and it is more common to work with modeled data the following parameter definitions focus on parameters derived from models.
It is often assumed that the maximum daily or hourly gust speed [m/s] at 10m height relates strongest to damage. The WMO defines a wind gust as the maximum of the wind averaged over 3 second intervals which is in most cases shorter than the model time step. Thus, many models rely on parametrization for gust speed. For example, the ECMWF Integrated Forecasting System deduces the magnitude of a gust within each time step from the time-step-averaged surface stress, surface friction, wind shear and stability. Other common parameters in use are daily or hourly mean or maximum wind speeds at 10m height which express the mean or maximum values of all model time steps in an hour or a day. The parametrized gust speed as well as mean wind speeds in a model grid cell can deviate widely from local observations." See lines 87-101.

**85) Reviewer comment:** L734: The agricultural sector is a crucial sector worldwide, not only in Europe.

**Our response:**

We changed the sentence as requested. See line 861.

**86) Reviewer comment:** L769-L778: I suggest deleting all information on climate indices that is not relevant for this review.

**Our response:**

We have followed the reviewer's suggestion and shortened this paragraph: "Whether or not wind-related agricultural damage will increase under continued warming is unclear. Peña-Angulo et al (2020) found that none of the five metrics linked to wind speed show a significant trend in either direction. However, the results are subject to considerable uncertainty given that convective events, which are associated with downbursts and straight-line winds, are poorly simulated in the current generation global circulation models." See lines 897-901.

**87) Reviewer comment:** L780: This section is also subdivided into subsection without numbering. Please homogenize the numbering throughout the entire text.

**Our response:**

We changed the numbering as requested.

**88) Reviewer comment:** L792: What kind of wind speed is mentioned here? Are these mean or instantaneous wind speed values? Please clarify.

**Our response:**

It is mean wind speed (typically 10 minute temporal resolution as also discussed in the text now), i.e. wind gusts will still be higher. We have changed the text accordingly: "The threshold values for mean wind speed […]". See lines 913 and 915.

**89) Reviewer comment:** L793: Of what kind are the mentioned "critical threshold values"? Do they correspond to CWS or vcrit or are they something different? Please clarify.

**Our response:**

We specified that we refer to CWS here (see line 916 in the revised version). We have also added that CWS is a standard term in forest ecology. See line 608 in the revised manuscript. Note also that we highlighted that v_crit and CWS are indeed similar, except the former is used in the context of building structures, whereas the latter in the context of trees (or forests for that matter). See lines 769-770 in the revised manuscript for clarification.

**90) Reviewer comment:** L805: Please check the citation formatting.

**Our response:**

Done.

**91) Reviewer comment:** P22L823: Is suggest deleting "extreme wind events". Or do you mean something other than storms? If so, please elaborate.

**Our response:**

We understand by extreme wind events a very high wind speed, but also a very high temporal and spatial variability of the wind vector. This includes events on different space-time scales (mesoscale to microscale): storms, tornadoes, vertical wind shear in low level jets, gustiness.

We changed this sentence to "Extreme wind events on different spatial and temporal scales, e. g. storms, gustiness or low-level jets, affect the energy production, the structural integrity and operational safety of wind turbines". See line 949-951.

**92) Reviewer comment:** L824: What do you mean by "stability"? Do you mean structural integrity? Please clarify.

**Our response:**

We changed the wording: "structural integrity and operational safety of the wind turbine." See lines 951.

**93) Reviewer comment:** L824-L826: What are "small-scale variations" in the wind field? Does this expression relate to the spatial wind speed pattern? Or does it also address the temporal wind field variability? Can "small-scale" be quantified? It does not sound to impact wind turbines a lot.

**Our response:**

We clarified the wording: small-scale does mean microscale.

"Microscale variability in the wind field occurs temporally (e.g., gustiness) and spatially (e.g., vertical wind shear). These variations of the wind field depend on the time of day and thus on the stability of the atmospheric stratification. There is also a dependence on the characteristics of the wind turbine site (land use, terrain). Microscale variations of the wind field influence both the wind potential and the operational reliability of a wind turbine." See lines 952-956.

**94) Reviewer comment:** L828-L834: Please rephrase these lines completely. The sequence of the contents is not stringent.

**Our response:**

Done. See lines 958-989.

**95) Reviewer comment:** L836: What is the meaning of "strong wind events" in this context? Does this refer to productive or destructive wind events? If it refers to productive wind events, then these events are an important part of the wind climate and should not be mentioned in this review.

**Our response:**

Productive wind events with high wind speeds result in high wind energy output. These events can also be associated with high wind shear, which causes high mechanical stress on the turbine and is thus problematic for operational safety or long-term structural integrity. A strong wind event in this sense includes both a productive and a destructive event.

We added to the text "strong wind events, which can be both includes both a productive and destructive" to clarify this point. See line 969.

**96) Reviewer comment:** L836-L843: How are these lines connected to the article's title? This is a list of wind indices that are related to wind turbine site assessment.

**Our response:**

Please see our response 95): Wind climatology and wind speed-based indices for site assessment include all possibly destructive wind events like a low-level jet which influence the long-term structural integrity. Production-based indices include these events too, but not all of these events, only events with a wind speed at hub height smaller than the cut-out wind speed at hub height.

**97) Reviewer comment:** L849: I suggest replacing "high-impact" in this context. High-impact or severe weather is normally weather that causes wide-spread damage. What could possibly be the "positive effect on wind energy production"? During high-impact weather wind turbines are shut down and the production rapidly decreases to zero.

**Our response:**

The cited work bears this title and refers to "Wind Energy Assessment during High-Impact Winter Storms in the Iberian Peninsula ". Gonçalves et al. (2020) stated: "… it is notable that the highest values of wind energy production occurred on the days of the storms' passage through the IP" (Iberian Peninsula). To resolve the contradiction in the choice of words, we have deleted the term "high-impact".

**98) Reviewer comment:** L867-L873: The definition of "compound event" is unclear. Can a compound event be related to a single hazard? If so, this should be made clear.

**Our response:**

The definition of compound is indeed quite wide. Considering the definition by Zscheischler et al. (2020):

"We organize the highly diverse compound event types according to four themes: preconditioned,where a weather-driven or climate-driven precondition aggravates the impacts of a hazard; multivariate, where multiple drivers and/or hazards lead to an impact; temporally compounding, where a succession of hazards leads to an impact; and spatially compounding, where hazards in multiple connected locations cause an aggregated impact."

To your specific question, yes, a single event (a windstorm) can cause strong winds and heavy precipitation (two different hazards). If the same area is affected – as is often the case – it is a compound event. We agree that the formulation was inaccurate and enhanced the text.

See lines 1003-1011.

**99) Reviewer comment:** L886: Please replace "Temperature" with the technically correct term "Air temperature". You mention "freezing of soil" which is directly related to "soil temperature".

**Our response:**

Done.

**100) Reviewer comment:** L897-L904: Your argument in these lines is not stringent. The statement "The impact of wind on drought is relatively small." cannot be backed by the statement "Wind is only included in some drought indices through evapotranspiration ..." This is not logical. This is a methodical aspect of wind impact assessment.

**Our response:**

We agree with the reviewer that the statement was inaccurate and was modified. See lines 1039-1048.

**101) Reviewer comment:** L936-P26L968: Why is only DWD mentioned here given the large number of national meteorological services? Given the scope of this review, it would be good to have more information from other weather services in the text as well.

**Our response:**

We include direct Links to exemplarily meteorological services (Sweden and Ireland) and also a link to Meteoalarm as combined effort of European Meteorological Services that provides an overview of European weather warnings. See line 1080.

**102) Reviewer comment:** L938-L940: More information about "some cases" would be useful. For example, how many and which weather services warn?

**Our response:**

We include a link to Meteoalarm in the text, where a list of the European weather services is given, that display warnings and possible impacts. See line 1090.

**103) Reviewer comment:** L942-L943: Previously, it was mentioned that the review was carried out for "five environments". Here, it is stated that threshold values were collected for "five sectors" and "five environments". The two expressions have different quality and scope. Which expression is correct? The correct expression should consistently be used throughout the text.

**Our response:**

We followed the suggestion and we use from now on only "sectors" in the entire manuscript.

**104) Reviewer comment:** L946: Here, the definitions for WL classes are provided. I suggest providing them at the beginning of the manuscript, in any case before L274.

**Our response:**

We agree. The information was moved to section 2.5. See lines 314-330.

**105) Reviewer comment:** L961: Please correct the formatting of "WL 1-4". What kind of "speeds" do you mean? Mean or instantaneous wind speed values? Please clarify.

**Our response:**

We corrected the formatting. We add (mean wind speed and wind gusts) to the caption. See lines 1119-1139.

**106) Reviewer comment:** L962-L968: Is it correct that "wind speed" and "gust speed" values are compared in this figure? What is the importance of this comparison?

**Our response:**

Indeed, this figure includes thresholds of wind gusts and mean wind speed in order to give an overview of the thresholds used to describe wind damage in the different sectors. Stakeholders themselves, belonging to just one sector, are often only interested in one variable or wind intensity, but the figure shows that in general the meteorological description and correct representation of mean wind and wind gusts from weak to strong intensities is important.

We changed the figure caption accordingly. See lines 1119-1139.

**107) Reviewer comment:** L962-L968: Do the gust speed values correspond to 3 s values?

**Our response:**

Yes, indeed. We state this now in the table caption.

**108) Reviewer comment:** L962-L968: What is the averaging interval of "mean wind"? Is the averaging interval for all displayed values the same?

**Our response:**

The averaging interval for the shown thresholds in the agriculture sector is 1h and in the energy sector as well. We state this now in the figure caption.

**109) Reviewer comment:** L962-L968: Is "mean wind" equivalent to "mean wind speed"?

**Our response:**

Yes, we changed to "mean wind speed". See line 1119.

**110) Reviewer comment:** L962-L968: Is the y-axis label "Wind speed" representative for all variables displayed in the figure?

**Our response:**

Yes, even if mean wind speeds and wind gusts (instantaneous) are shown in the figure, all threshold values refer to the variable wind, with the physical unit m/s. We state this now in the figure caption.

**111) Reviewer comment:** L962-L968: In the figure the information "Wind in hub height" is provided. Does this correspond to "Wind speed in hub height"? In the figure caption "wind speed measured at the height of the wind turbines" is mentioned. Is this the same quantity? Please clarify.

**Our response:**

Yes, it is the same quantity. In the figure, a shorter description was used to save space. For clarification, the caption has been adjusted.

**112) Reviewer comment:** L962-L968: I suggest replacing "Critical thresholds" with "Critical threshold ranges". This seems to be shown by this figure.

**Our response:**

Done.

**113) Reviewer comment:** L972: Please uppercase "central Europe".

**Our response:**

Done.

**114) Reviewer comment:** L997-L1007: Please specify "indices" at every occurrence. Which indices do you mean? Wind indices? Storm indices? Storm damage indices?

**Our response:**

We specified as requested, that these indices were topographic indices. See line 1172.

**115) Reviewer comment:** L1004: Please define the abbreviations R2 and AUC.

**Our response:**

After rewriting this part of the manuscript we do not use R2 and AUC any longer.

---

## Author Comment (AC2)

Reviewer #2

Thank you very much for reviewing our manuscript and your helpful comments.

**1) Reviewer comment:** The authors provide an overview of (1) the meteorological phenomenon wind (2) its processes in interacting with the surface from a physical as well as an impact perspective and (3) a large collection of indices that are structured based on five environments: forests, urban, transport, agriculture, and wind-based energy production. These environments represent different communities in scientific literature as well as different sectors of socio-economic impacts. The authors provide a synthesis, an outlook and discuss open research questions.

**Our response:**

Thank you for your kind remarks.

**2) Reviewer comment:** The manuscript would greatly benefit, if the different environments could be more synthesized in the outlook sections. Many open research questions seem similar in the different communities and could be tackled synergistically in the future. Additionally more explicitly spelling out some generalized conclusions about the different indices in the different environments before the outlook would increase the usefulness of this review.

**Our response:**

That is a good suggestion. We added several specific research points to the outlook section.

See lines 1194-1275.

**3) Reviewer comment:** This manuscript features damage, impact and risk throughout the manuscript, but it mentions the socio-economic literature community (especially regarding exposure and vulnerability) mainly when such elements are used in indices in section 4. Maybe it would be beneficial to more often link to this body of research also in other sections:

**Our response:**

We tried to do this, we added some more references to the introduction, but the most appropriate place for the socio-economic literature is in section 4. So some imbalance is unavoidable.

See lines 85, 340 and 1149 where we added some socio-economic literature.

**4) Reviewer comment:** General introduction: How does this review position itself compared to reviews over different sectors in the impact modelling community (e.g. Merz et al. 2020 for windstorms and severe convective storms)?

**Our response:**

The focus of our review paper is quite different than Merz et al. 2020, which focusses on the forecasting of impacts related with many natural hazards, and not only wind(storms). They also consider only a few indices, where we discuss many more. We have added a sentence in the introduction to specify the different focus of Merz et al.

See lines 78-80.

**5) Reviewer comment:** L867ff/L1009ff How do the non-climatic drivers related to vulnerability and exposure mentioned in Zscheischler et al. (2018) play into the development of compounded indices?

**Our response:**

To our best knowledge non-climate drivers (like the mentioned vulnerability and exposure) can be considered for the development of compound indices. However, this is not typically the case. We state now that an important challenge is the inclusion of non-climate drivers in the indices. See lines 1003-1011.

**6) Reviewer comment:** L1: is the word "damage" broad enough? The manuscripts also mentions positive or indirect effects of wind. Why not mention the word "indices" in the title?

**Our response:**

We added "indices" to the title and define "damage" in the first paragraph of the introduction to ensure that its meaning is broad enough. See lines 61-62.

**7) Reviewer comment:** L39: "Fortunately, simple indices and thresholds are as effective as complex mechanistic models for many applications." The "complex mechanistic models" are only mentioned in comparison with indexes but never fully defined, This term should be defined somewhere in the manuscript (e.g. in section 3).

**Our response:**

We added the last paragraph in "3.1. The physics of fine scale interactions between surfaces and wind". See lines 448-457.

**8) Reviewer comment:** L40: "Nonetheless, the multitude of indices and thresholds available requires a careful selection process according to the target environment". This "careful selection process" could be taken up and expanded upon with useful suggestions at the end of the manuscript e.g. after L1001.

**Our response:**

We followed the request of the reviewer and added the text in the suggested location. See lines 1169-1193.

**9) Reviewer comment:** L78: It would be important if the manuscript would include information about the applied methodology that lead to this manuscript, here is just one possible location in the text: From the acknowledgement, I assume that a group of experts formed in the project ClimXtreme. The selection of the papers and their categorization in this review is an outcome of many discussions or workshops within this group and of individual expert knowledge. If this is not the case: how where the studied papers selected? Where there any relevant decisions what overlapping/neighboring fields of literature to include or exclude (e.g. other types of indices, other environments)?

**Our response:**

We added the text as requested. Lines 109-117.

**10) Reviewer comment:** L623: shouldn't Koks and Haer (2020, already in References) also be mentioned here as an example of loss models

**Our response:**

We added the citation.

**11) Reviewer comment:** L940: National meteorological services do not only indicate the possible consequences, but take the consequences and the probabilities of these consequences as input into their warning decision (e.g. Neal et al. 2014) or they plan to do so in the future (Kaltenberger et al. 2020).

**Our response:**

Thank you for the advice. This is a helpful addition and we have added it to the manuscript. See lines 1086-1092.

**12) Reviewer comment:** L942ff: This paragraph could be structured and phrased more clearly. It would also be helpful to include the references for the thresholds of the different environments in the main text and not only in the supplementary material.

**Our response:**

We add line breaks and some of the references from the supplementary material. See lines 1094-1117.

**13) Reviewer comment:** L946: It is unclear how reaching a critical warning threshold in wind speed is related to the spatial extent. Mainly, it is unclear if the threshold is applied to each location (as it

normally is for warnings) or once per weather phenomena (e.g. for the maximum wind speed over the whole affected area of an event similar to a storm severity index). If it is applied to each location, can't a larger area (e.g. national area) have reached WL2? If it is about a localized damage having consequences for society on a larger spatial scale, then this needs to be said more clearly.

**Our response:**

The damage localization belongs to the consequences for society. Therefore we name the category National "Impact". We add the word "impact" to regional and local as well, to clear this up. See lines 1101-1117.

**14) Reviewer comment:** L961: The names of the different threshold ranges (e.g. local, regional, cut-in speed, cut- out speed) are only understandable using the supplementary material (S2). It would be better if these names would be explained in the caption or at least the previous paragraph of the main text.

**Our response:**

We add short explanation of the threshold ranges in the figure caption.

**15) Reviewer comment:** L1001: "Such a methodology needs to be developed on a large spatial scale to evaluate in which regions certain groups of indices are useful." It would be nice if this sentence would be expanded so its meaning is made clearer. Also what else is needed to allow such evaluations on a large spatial scale? Could the "careful selection process" mentioned in the abstract be expanded on here?

**Our response:**

We expanded and clarified this section as requested. Furthermore, the statement "careful selection process" is explained now here. See lines 1169-1193.

**16) Reviewer comment:** L1002ff: Data on "given metrics" are often scarcely available, if the "given metric" is related to a socio-economic impact. This should be mentioned.

**Our response:**

We changed the text as requested. See lines 1169-1193.

**17) Reviewer comment:** L1002ff: what about other possible solutions? E.g. the inclusion of user preference or expert knowledge in the development of indices using co-design (e.g. Gebhardt et al. 2019 cited in Merz et al. 2020)

**Our response:**

We thank the reviewer for the suggestion, and we incorporated it in the text. See lines 1174-1184.

**18) Reviewer comment:** L1016-1039: It would increase the usefulness of the manuscript if outlook and open research questions could be unified over the five environments. Would it be possible to combine these two paragraphs or to the split according to common questions? Surely, not only the forest setting is lacking damage data etc.

**Our response:**

We significantly extended the outlook section, devoting a sub-section for each of the sectors. See lines 1194-1279.

**19) Reviewer comment:** L1021-1030 and L1034-1038: In my understanding, better knowledge of the spatial variability of the environments (e.g. forests or urban) is important for two reasons: (1) it has an effect on the small-scale interactions of the wind field with the surface (2) it informs difference in vulnerability and spatial distribution (e.g. of the value) of the impacted entity (e.g. trees and buildings). These two reasons could be more clearly distinguished in this paragraph but also in the section 4.

**Our response:**

We added these key aspects in all sections of the outlook and especially for forest, urban and agriculture sectors.